# A Comprehensive Study on Text-attributed Graphs: Benchmarking and Rethinking

**Hao Yan[1]**\*, **Chaozhuo Li[2]**\*, **Ruosong Long[3], Chao Yan[4], Jianan Zhao[6], Wenwen Zhuang[1]**
**Jun Yin[1], Peiyan Zhang[5], Weihao Han[7], Hao Sun[7], Weiwei Deng[7], Qi Zhang[7],**
**Lichao Sun[8], Xing Xie[2], Senzhang Wang[1]**†
[1]Central South University, [2]Microsoft Research Asia, [3]University of Birmingham
[4]Peking University, [5]Hong Kong University of Science and Technology
[6]Université de Montréal, [7]Microsoft, [8]Lehigh University

## Abstract

Text-attributed graphs (TAGs) are prevalent in various real-world scenarios, where each node is associated with a text description. The cornerstone of representation learning on TAGs lies in the seamless integration of textual semantics within individual nodes and the topological connections across nodes. Recent advancements in pre-trained language models (PLMs) and graph neural networks (GNNs) have facilitated effective learning on TAGs, garnering increased research interest. However, the absence of meaningful benchmark datasets and standardized evaluation procedures for TAGs has impeded progress in this field. In this paper, we propose CS-TAG, a comprehensive and diverse collection of challenging benchmark datasets for TAGs. The CS-TAG datasets are notably large in scale and encompass a wide range of domains, spanning from citation networks to purchase graphs. In addition to building the datasets, we conduct extensive benchmark experiments over CS-TAG with various learning paradigms, including PLMs, GNNs, PLM-GNN co-training methods, and the proposed novel topological pre-training of language models. In a nutshell, we provide an overview of the CS-TAG datasets, standardized evaluation procedures, and present baseline experiments. The entire CS-TAG project is publicly accessible at `https://github.com/sktsherlock/TAG-Benchmark`.

## 1 Introduction

Graphs are ubiquitous in modeling the relational and structural aspects of real-world objects across various domains, such as social networks, transportation system networks, and biological protein-protein networks [1, 2, 3, 4]. In many real-world graphs, nodes are often associated with text attributes, giving rise to the text-attributed graphs (TAGs) [5, 6]. TAGs are prevalent in various scenarios, such as social graphs where each user is accompanied by a textual description and paper citation graphs where textual content is linked to each respective paper [7, 8]. The exploration of learning methodologies applied to TAGs has emerged as a prominent research area within multiple fields, including graph learning, information retrieval, and natural language processing [9].

The nucleus of learning on TAGs lies in the effective integration of both the node attributes (textual semantics) and graph topology (structural connections) to facilitate the learning of node representations. The textual information associated with each node offers a wealth of semantic content, enabling the characterization of individual node properties, which could be captured by the pre-trained language models (PLMs) [10, 11, 12, 13, 14]. Meanwhile, the structural information encoded

---

\*Equal contribution and co-first authors.
†Corresponding author.

37th Conference on Neural Information Processing Systems (NeurIPS 2023) Track on Datasets and Benchmarks.

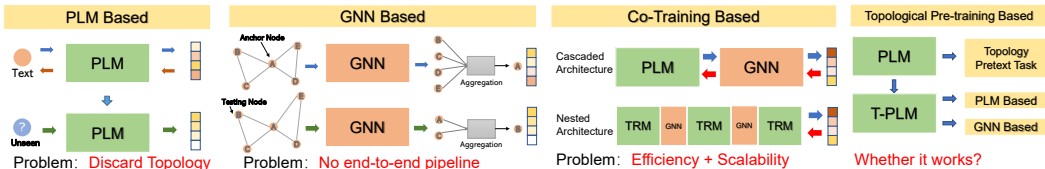

Figure 1: The traditional text attributed graph representation learning pipeline.

within the graph topology presents the inherent proximity relationships between nodes. Graph neural networks (GNNs) have proven to be effective in capturing such structural relations based on the message-passing mechanism [7, 15, 16, 17, 18, 19, 20, 21, 22].

PLM-based and GNN-based methods are two prevalent types of learning paradigms on TAGs as illustrated in Figure 1. PLM-based methods generally input the textual content derived from the target node into a pre-trained language model. However, the knowledge of topology resulting from the high non-linearity of graph structure within TAGs is largely discarded by the PLM-based methods [9]. Conversely, GNN-based methods are capable of preserving the intricate graph topology information with greater fidelity. Nevertheless, an inherent limitation plaguing GNN-based methods lies in the disconnected modeling process of node attributes and graph topology. Specifically, most GNNs pre-model node attributes as static representations, treating them as fixed and unlearnable parameters during the message passing process. Consequently, the gradients stemming from the learning objective of GNNs cannot be effectively back-propagated into the attribute modeling. This discrepancy in the training procedure hinders the attainment of an optimal solution, as it fails to guarantee end-to-end training, thereby impeding the overall effectiveness of the approach.

To simultaneously enjoy the merits of GNNs and LMs, several recent endeavors shed light on the co-training paradigm as shown in Figure 1. LMs and GNNs are combined in a cascaded [23, 24, 25] or nested [5] manner, establishing a unified end-to-end training paradigm to model the node attributes and graph topology jointly. Despite its theoretical appeal, the co-training method suffers from severe scalability issues as its memory complexity is proportional to the graph size as neighborhood texts are also encoded [9]. Motivated by the recent advancements in pre-training techniques, a novel inquiry emerges: *Can we pre-train the language models to understand the graph topology?* If we can effectively encode topological information into LMs through appropriate pre-training tasks, LMs could serve as the foundational model for learning on TAGs. Topological pre-trained LMs eliminate the explicit GNN aggregations, thereby circumventing the efficiency challenges encountered in the co-training paradigm. However, the design of suitable and effective pre-training tasks to encode valuable knowledge derived from intricate graph topology into LMs remains an open question.

In order to delve deeply into the intricate interplay between textual semantics and graph topology within TAGs, we embark on an unprecedented exploration to investigate the optimal training paradigm for various TAGs. Existing text-attributed graph datasets (e.g., Cora [8], WikiCS [26], Amazon-Photo [27]) cannot meet our requirements, as they solely offer node attribute embeddings, devoid of the original textual sentences. To overcome this limitation, we meticulously curate a novel and comprehensive dataset, dubbed CS-TAG, comprising eight distinct TAGs sourced from diverse domains. This carefully crafted dataset serves as a solid foundation for future research endeavors, facilitating in-depth investigations in this burgeoning field. Moreover, extensive experiments are conducted on the CS-TAG dataset to provide comprehensive and reliable benchmarks. All aforementioned learning paradigms are thoroughly evaluated and analyzed. Experimental results and detailed discussions further reveal the underlying correlations between graph topology and textual attributes, drawing deep insight into the inherent characteristics of the TAGs. Our contributions are summarized as follows:

1. To the best of our knowledge, CS-TAG is the first open dataset specifically designed for text-attributed graphs. TAGs from a variety of fields are collected, cleaned, and organized as the final structured dataset. We provide researchers with original links and data cleaning codes to facilitate their access and reprocessing of these datasets in accordance with their research interests and requirements. The entire CS-TAG project is publicly accessible as an open source repository on Github, accessible at `https://github.com/sktsherlock/TAG-Benchmark`.

2. In contrast to previous topology-driven graph learning models, our work underscores the vital significance of deep node attribute modeling. This novel perspective sheds light on the design of next-generation GNNs by emphasizing the incorporation of deep node attribute understanding.

3. We investigate the novel problem of topological pre-training of language models, aiming at teaching LMs to understand topological structures. This innovative training paradigm exhibits remarkable performance on the CS-TAG dataset in terms of effectiveness and efficiency, which contributes to broadening the scope of language model pre-training.

4. Extensive experiments are conducted across eight diverse datasets, focusing on two downstream tasks: node classification and link prediction. Such experiments serve as a rigorous evaluation of various learning paradigms, providing precise and dependable benchmarks for future endeavors.

## 2 Related Work

In this section, we first briefly introduce three popular learning paradigms for TAGs. After that, the comparisons between the existing graph learning benchmarks and the proposed CS-TAG are also discussed. Refer to Appendix A for more detailed reviews of the related models. We have implemented most of the algorithms discussed in this section in the repository.

**PLM-based methods.** The PLMs refer to universal language models that possess enhanced semantic understanding due to their pre-training on a vast corpus [28]. The early works on modeling textual attributes were based on shallow networks, e.g., Skip-Gram[29] and GloVe[30]. In recent years, the backbone networks dominated by the pre-training-fine-tuning paradigm are rapidly scaling up: from ELMo[31], GPT[32], to BERT [10], RoBERTa [12], DeBERTa [13]. The large-scale models, which get fully trained with massive data, demonstrate superior performances on general NLP tasks. One of the most critical usages of PLMs is text representation, where the underlying semantics of texts are captured by low-dimensional embeddings. On the TAGs, the PLMs use the local textual information of each node to learn a good representation for the downstream task [9].

**GNN-based methods.** As graph representation learning enjoys explosive growth in machine learning, numerous research works have been proposed for various tasks including node classification [15], link prediction [21], and so on. Graph neural networks are recognized as powerful tools for modeling graph data. Such methods (e.g., GCN [15], GAT [16], GraphSAGE [7], GIN [17], RevGAT [33]) learn effective message-passing mechanisms such that information between the nodes can get aggregated for expressive graph representations. GNNs generally adopt the "cascade architecture" suggested by GraphSAGE for textual graph representation: node features are encoded independently using text modeling tools (e.g. PLMs) and subsequently aggregated by GNNs to produce the final representation.

**Co-training methods.** The aforementioned two types of paradigms primarily focus on modeling partial information, which limits their ability to learn comprehensive features. Several recent endeavors propose to co-train GNNs and LMs to enjoy the merits from both sides. Specifically, LMs and GNNs are combined in the cascaded [24] or nested manner [5]. The outputs generated by LMs serve as inputs for GNNs, and vice versa. The parameters of both LM and GNN are updated through the back-propagation of gradients from downstream tasks. However, this co-training paradigm suffers from serious scalability problems, as all neighbors need to be encoded from scratch by the LMs, incurring significant additional computational costs [9].

**Benchmarks for graph representation learning.** Several established graph benchmarks have been developed and widely adopted [34, 35, 27, 36]. However, when it comes to learning on TAGs, these benchmarks exhibit notable deficiencies. Firstly, a majority of these datasets suffer from the absence of raw textual information, limiting the investigation of attribute modeling's effectiveness. Secondly, these datasets often neglect to explore the impact of text attribute modeling on GNNs. Thirdly, these datasets are predominantly small in scale. Thus, there is a compelling necessity to construct a comprehensive large-scale dataset for TAGs.

## 3 CS-TAG: A Comprehensive Dataset and Benchmark for TAGs

In this section, we commence by providing a concise summary of the constructed CS-TAG benchmark in Section 3.1. Subsequently, we present the details of the construction of CS-TAG in Section 3.2, including data collection, cleaning, and labeling. Moreover, we elucidate the details of GNN-based, PLM-based, and Co-training learning paradigms in Section 3.3. Finally, the proposed topological pre-training of LMs is presented in Section 3.4.

### 3.1 Overview of CS-TAG

In order to address the limitations inherent in prior researches, we propose the establishment of the text-attributed graph benchmark, dubbed *CS-TAG*, which serves as a standardized evaluation framework for assessing the efficacy of representation learning techniques on TAGs. To ensure scalability, CS-TAG includes datasets of varying sizes and incorporates scalable baselines consisting of PLMs, GNNs, and co-training methods. This enables researchers to evaluate the performance of their models across a broad range of dataset scales. To enhance usability, we provide a modular pipeline that simplifies the implementation of different models within the CS-TAG. Such a modular architecture enables researchers to easily integrate their novel methods and compare them with existing approaches. In addition, we are committed to maintaining a public leaderboard for TAGs, serving as a repository for the latest advancements in the field. This platform will continuously update text-attributed graph datasets that possess practical and research value, fostering ongoing progress and collaboration within the community. Overall, CS-TAG serves as a scalable, unified, modular, and consistently updated evaluation framework for assessing the performance of representation learning methods on text-attributed graphs.

### 3.2 Dataset Construction

In order to thoroughly investigate the performance of different learning paradigms on TAGs, we conduct an extensive survey of various text-attributed graph datasets that have been previously utilized in the literature. Our observations reveal that many commonly employed node-level datasets are essentially text-attributed graphs. For instance, well-known citation graphs such as Cora, PubMed, Citeseer [8], and ogbn-arxiv [34] are all TAGs. These datasets derive node attributes from textual information, such as the title and abstract of papers. Additionally, academic collaboration networks such as Coauthor CS/Physics [27] set node attributes derived from keywords defined in the papers.

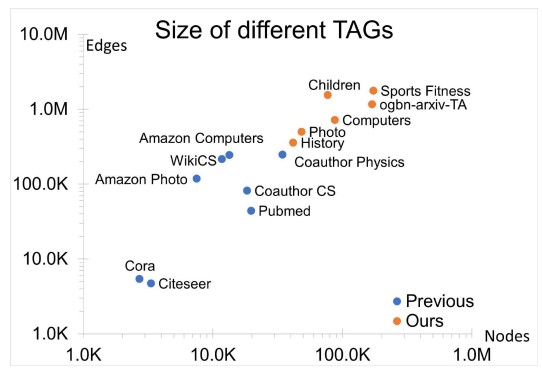

Figure 2: The differences between the TAGs datasets in CS-TAG (used for node classification) and the previous datasets.

However, while these datasets are frequently employed by GNNs, they possess obvious inadequacies when exploring representation learning on TAGs. Firstly, most of these datasets lack the availability of raw textual information, bringing challenges to investigating the effectiveness of attribute modeling on these datasets. Secondly, these datasets generally overlook the exploration of text attribute modeling's impact on GNNs. A majority of these datasets employ simplistic bag-of-words models or traditional text encoding techniques like GloVe or Skip-Gram to represent text attributes, which are kind of outdated. Lastly, these datasets are predominantly small in scale, leading to a lack of differentiation between different learning models across numerous datasets.

To address these limitations, we have taken proactive steps to collect and construct some novel datasets of TAGs. Figure 2 illustrates the number of nodes/edges in the previous datasets and the proposed CS-TAG. One can clearly see that the TAGs within CS-TAG are comparatively larger than the counterparts. Here, we present the details of shopping graphs as an example. We extract datasets from the Amazon dataset [37], including Books-Children/History, Ele-Computers/Photo, and Sports-Fitness. Nodes represent different types of items, while edges indicate items that are frequently purchased or browsed together. Node labels are assigned based on the product category. To explore the influence of attributes in the text-attribute graphs, distinct text attributes have been provided for each of these datasets. For example, in the Books-Children/History dataset, node attributes are derived from the title and description of the respective books, such as "Description: Collection of Poetry; Title: The golden treasury of poetry". The Sports-Fitness dataset only incorporates node attributes from the title of the sports items, such as "Girls Ballet Tutu Neon Orange". In the Ele-Computers/Photo dataset, node attributes are obtained from high-rated reviews and product summaries, for instance,

Table 1: Statistics of text-attributed graph dataset used in CS-TAG.

| | Dataset | Nodes | Edges | Class | Domain | Modeling | Scale | Tasks | Raw Text |
|---|---|---|---|---|---|---|---|---|---|
| Previous | WikiCS | 11,701 | 216,123 | 10 | Wikipedia | GloVe | Medium | Node classification | ✗ |
| | Cora | 2,708 | 5,429 | 7 | Academic | Bag of words | Small | Node classification | ✗ |
| | Citeseer | 3,327 | 4,732 | 6 | Academic | Bag of words | Small | Node classification | ✗ |
| | Pubmed | 19,717 | 44,338 | 3 | Academic | Bag of words | Medium | Node classification | ✗ |
| | ogbn-arxiv | 169,343 | 1,166,243 | 40 | Academic | Skip-Gram | Large | Node classification | ✔ |
| | Coauthor CS | 18,333 | 81,894 | 15 | Academic | Bag of words | Medium | Node classification | ✗ |
| | Coauthor Physics | 34,493 | 247,962 | 5 | Academic | Bag of words | Medium | Node classification | ✗ |
| | Amazon Photo | 7,487 | 119,043 | 10 | E-commerce | Bag of words | Small | Node classification | ✗ |
| | Amazon Computers | 13,381 | 245,778 | 8 | E-commerce | Bag of words | Medium | Node classification | ✗ |
| Ours | ogbn-arxiv-TA | 169,343 | 1,166,243 | 40 | Academic | PLMs | Large | Node classification | ✔ |
| | Books-Children | 76,875 | 1,554,578 | 24 | E-commerce | PLMs | Large | Node classification | ✔ |
| | Books-History | 41,551 | 358,574 | 12 | E-commerce | PLMs | Large | Node classification | ✔ |
| | Ele-Computers | 87,229 | 721,081 | 10 | E-commerce | PLMs | Large | Node classification | ✔ |
| | Ele-Photo | 48,362 | 500,928 | 12 | E-commerce | PLMs | Large | Node classification | ✔ |
| | Sports-Fitness | 173,055 | 1,773,500 | 13 | E-commerce | PLMs | Large | Node classification | ✔ |
| | CitationV8 | 1,106,759 | 6,120,897 | - | Academic | PLMs | Large | Link Prediction | ✔ |
| | GoodReads | 676,084 | 8,582,324 | - | E-commerce | PLMs | Large | Link Prediction | ✔ |

"Great camera for the price! This camera takes crystal clear photos and is cheap too!". Further details on the dataset construction process can be found in Appendix B.

Additionally, we construct two other datasets, CitationV8 and GoodReads, for the link prediction task. The CitationV8 dataset represents a citation network extracted from DBLP [38]. Node attributes in CitationV8 are derived from the titles and abstracts of research papers. Each edge signifies a citation relationship between two papers. The GoodReads dataset, on the other hand, originates from a prominent book review website.[3] This dataset captures the "similar item" linking relationship between books and provides valuable information about the attributes of each book, such as the title and description. Therefore, we leverage the GoodReads dataset to construct link prediction tasks, which involve predicting relationships between similar books. Detailed descriptions of all the aforementioned datasets can be found in Appendix B.

## 3.3 Conventional Learning Paradigms on TAGs

Existing learning paradigms on TAGs can be broadly classified into three distinct categories: 1) GNN-based methods: These methods primarily leverage GNNs as the foundational model for capturing the underlying graph topology structures through message-passing mechanisms. 2) PLM-based methods: These approaches rely on prevalent pre-trained language models to capture the semantics from the textual node attributes, which excel in their ability to comprehend text semantics and exhibit strong transferability. 3) Co-training methods: This paradigm involves the joint learning of GNNs and LMs under a unified framework [23] to enjoy the merits from both sides. Next, we will give the formulaic definitions of these three paradigms.

**GNN-based Paradigm.** GNNs are employed to propagate information across the graph nodes, allowing for the extraction of meaningful representations via message passing, which are formally defined as follows:

$$\boldsymbol{h}_u^{(k+1)} = \text{UPDATE}_{\boldsymbol{\omega}}^{(k)}\left(\boldsymbol{h}_u^{(k)}, \text{AGGREGATE}_{\boldsymbol{\omega}}^{(k)}\left(\left\{\boldsymbol{h}_v^{(k)}, v \in \mathcal{N}(u)\right\}\right)\right) \tag{1}$$

where $k$ represents the layers of GNNs, $\mathcal{N}$ denotes the set of neighbors, $u$ denotes the target node, $\boldsymbol{\omega}$ means the learning parameters in GNNs. Please note that, the initial node feature vector $\boldsymbol{h}_u^{(0)}$ using pre-learned by PLMs or other shallow text encoder (e.g., Skip-Gram). Such attribute modeling phase is performed independently of the subsequent training of GNNs. Gradients from the GNN training objectives are unable to be back-propagated into the PLMs to update their parameters. And this decoupling of PLMs and GNNs impedes the overall effectiveness.

**PLM-based Paradigm.** PLM-based methods leverage the effectiveness of pre-training techniques to enhance the modeling of text within each node. The formulation of these methods is as follows:

$$\boldsymbol{h}_u^{(k+1)} = \text{UPDATE}_{\boldsymbol{\psi}}^{(k)}\left(\boldsymbol{h}_u^{(k)}\right) \tag{2}$$

where $\boldsymbol{\psi}$ denotes the learnable parameters in PLMs. PLMs advance the modeling of node text attributes. However, incorporating crucial topological context into PLM-based paradigms remains a challenge, particularly when the available textual data is limited.

---

[3]https://www.goodreads.com/

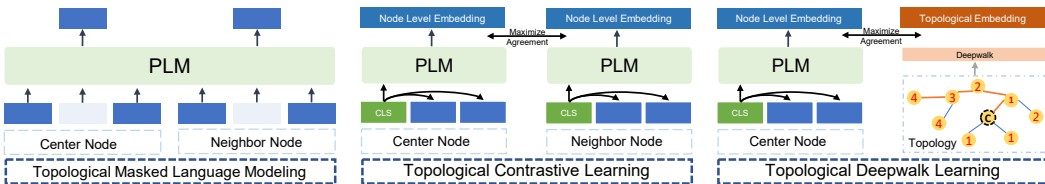

Figure 3: Illustrations of different topological pre-training methods.

**Co-training Paradigm.** GNNs and LMs are jointly trained under a unified training framework:

$$f_{\boldsymbol{\Theta}}(A, T) = \text{GNN}_{\boldsymbol{\omega}}\left(A, \text{PLM}_{\boldsymbol{\psi}}(T)\right), \boldsymbol{\Theta} = \{\boldsymbol{\omega}, \boldsymbol{\psi}\} \tag{3}$$

where $f$ denotes the learning function and $\boldsymbol{\Theta}$ denotes the entire learnable parameters, which are derived from both the GNN and PLM modules. The outputs generated by the LM serve as input to the GNN. Notably, the gradients obtained from the GNN can be back-propagated to the LM, enabling the update of its parameters. However, the co-training method faces significant scalability challenges. This is primarily due to the memory complexity associated with encoding neighborhood texts, resulting in a memory requirement that scales linearly with the size of the graph [9].

### 3.4 Topological Pre-training of Language Models

The incorporation of explicit GNN aggregations in the co-training paradigm introduces inherent challenges in terms of training complexity and resource requirements. This is primarily due to the simultaneous modeling of texts from both the center node and its neighbors. Therefore, this brings us to a question: *is there a training paradigm to enjoy the merits of graph topology while avoiding the explicit GNN operations?* Inspired by the recent advancements in pre-training techniques [39, 40], our motivation lies in teaching language models to understand the topological structures. Three topological pre-training tasks are proposed to impart graph structures into the LMs, enabling them to better comprehend and capture the underlying topology.

**Topological Masked Language Model (TMLM).** Inspired by the task of masked language modeling, we propose a novel topological masked language model (TMLM) to capture the first-order connections on the token level. Given a center node $c$ and one of its neighbors $n$, their corresponding text is formally defined as $\boldsymbol{T}^{(c)} = \{t_1^{(c)}, t_2^{(c)}, \ldots, t_k^{(c)}\}$ and $\boldsymbol{T}^{(n)} = \{t_1^{(n)}, t_2^{(n)}, \ldots, t_u^{(n)}\}$, respectively. We randomly replace a subset of tokens in the center node $\boldsymbol{T}^{(c)}$ and $\boldsymbol{T}^{(n)}$ with a special token [MASK]. The objective of TMLM is to predict the masked tokens. Let $\boldsymbol{\Phi}^{(s)} = \{\phi_1^{(c)}, \phi_2^{(c)}, ..., \phi_{m-1}^{(n)}, \phi_m^{(n)}\}$ represents the indexes of the $m$ masked tokens in the sentence $\boldsymbol{T}^{(c)}$ and $\boldsymbol{T}^{(n)}$. Let $\boldsymbol{T}_{\Phi}^{(s)}$ denote the set of masked tokens in $\boldsymbol{T}^{(c)}$ and $\boldsymbol{T}^{(n)}$, and $\boldsymbol{T}_{-\Phi}^{(s)}$ denote the set of observed (unmasked) tokens. The objective of TMLM is:

$$\mathcal{L}_{\text{tmlm}}(\boldsymbol{T}_{\Phi}^{(s)}|\boldsymbol{T}_{-\Phi}^{(s)}) = \frac{1}{m}\sum_{i=1}^{m}\log p(t_{\phi_i}|\boldsymbol{T}_{-\Phi}^{(s)}; \theta). \tag{4}$$

in which $\theta$ denotes the learnable parameters.

**Topological Contrastive Learning (TCL).** Inspired by contrastive learning [41, 42, 43, 44], we propose a novel topological contrastive learning (TCL) task to capture the first-order topological information in the node level. Given a center node $c$ and one of its neighbors $n$, their corresponding node-level (sentence/document-level) embedding is derived from their CLS token in the $\boldsymbol{T}_{\Phi}^{(c)}$ and can be formally defined as $\boldsymbol{h}^c, \boldsymbol{h}^n$. The objective of TCL is to bring the center node $\boldsymbol{h}^c$ closer to its neighbors $\boldsymbol{h}^n$ while pushing itself farther away from other nodes. Denoting the cosine similarity function as $\text{sim}(\boldsymbol{h}^c, \boldsymbol{h}^n) = \boldsymbol{h}^{c\top}\boldsymbol{h}^n/\|\boldsymbol{h}^c\|\|\boldsymbol{h}^n\|$. The objective of TCL is:

$$\mathcal{L}_{\text{tcl}} = -\log \frac{\exp(\text{sim}(\boldsymbol{h}^c, \boldsymbol{h}^n)/\tau)}{\sum_{n'=1, n'\neq n}^{N}\exp(\text{sim}(\boldsymbol{h}^c, \boldsymbol{h}^{n'})/\tau)}, \tag{5}$$

where $\tau$ denotes the temperature parameter, $N$ denotes the batch size.

**Topological Deepwalk Learning (TDK).** TMLM and TCL mainly capture low-order structural information, while the higher-order structural information still needs to be captured by designing

Table 2: Accuracy comparison among GNNs on the ogbn-arxiv-TA within different PLMs' node features. "Scale" means the different versions of PLMs (number of parameters). "Diff" denotes the performance gap between the best and worst performers. We mark the best performer in each row with blue bold font and mark the best performer in each column with black bold font.

| Scale | PLMs | Arxiv | | | | | | | | | |
|-------|------|-----|-----|------|--------|---------|-----|------|-------|------|------|
| | | GCN | GAT | SAGE | RevGAT | NFormer | GIN | JKNet | APPNP | MoNet | MLP |
| Small | BERT-Tiny | 72.03 | 72.25 | 72.35 | **72.52** | 71.91 | 68.42 | 69.50 | 71.63 | 45.13 | 57.22 |
| | ELECTRA | 68.45 | 70.97 | 69.63 | **71.12** | 69.45 | 58.09 | 62.87 | 59.55 | 36.65 | 36.58 |
| | DistilBERT | 73.39 | 73.48 | 74.48 | **74.68** | 73.56 | 72.30 | 71.44 | 74.01 | 50.51 | 68.11 |
| Base | ELECTRA | 70.81 | 71.67 | 70.82 | **71.96** | 70.43 | 64.88 | 63.41 | 65.62 | 38.91 | 48.56 |
| | BERT | 73.30 | 73.40 | 74.14 | **74.59** | 72.80 | 71.94 | 70.08 | 73.90 | 46.90 | 67.35 |
| | RoBERTa | 73.56 | 73.38 | 74.52 | **74.82** | 73.12 | 72.63 | 69.40 | 74.01 | 44.53 | 69.31 |
| | DeBERTa | 68.15 | 66.56 | 67.58 | **68.26** | 67.11 | 62.05 | 44.16 | 52.37 | 29.67 | 47.07 |
| Large | ELECTRA | 70.44 | 71.01 | 70.72 | **72.56** | 70.04 | 64.47 | 58.34 | 64.52 | 37.26 | 47.72 |
| | BERT | 73.25 | 73.37 | 74.15 | **74.68** | 73.12 | 71.88 | 68.70 | 73.53 | 43.31 | 66.85 |
| | RoBERTa | 73.95 | 73.72 | 74.64 | **74.99** | 73.12 | 73.10 | 68.10 | 74.17 | 44.01 | 69.51 |
| | DeBERTa | 72.57 | 71.50 | 73.22 | **73.59** | 71.88 | 71.25 | 54.41 | 69.28 | 33.53 | 66.28 |
| | **Diff** | 5.80 | 7.16 | 7.06 | 7.54 | 6.45 | 15.01 | 27.28 | 21.80 | 20.84 | 32.93 |

suitable tasks. Considering that algorithms like Deepwalk [45] can capture higher-order structural information in the graph, we try to use the node representations learned by Deepwalk to augment the representations learned by LM. We first feed the whole graph structure into Deepwalk to get the corresponding representation $\boldsymbol{k}^c$ of each node $c$. The objective of TDK is to bring the center node $\boldsymbol{h}^c$ closer to its representation learned from Deepwalk $\boldsymbol{k}^c$. The objective of TDK is:

$$\mathcal{L}_{\text{tdk}} = -\log \frac{\exp(\text{sim}(\boldsymbol{h}^c, \boldsymbol{k}^c)/\tau)}{\sum_{c'=1, c' \neq c}^{N} \exp(\text{sim}(\boldsymbol{h}^c, \boldsymbol{k}^{c'})/\tau)}, \tag{6}$$

## 4 Experiments

**Baselines.** **(1)** For GNN-based methods, we select 9 popular GNN models: GCN [15], GAT [16], GraphSAGE [7], RevGAT [33], NodeFormer [46], GIN [17], JKNet [18], MoNet [47] and APPNP [48]. **(2)** For PLM-based methods, we select 5 different PLMs with different scales: a) Small parameter scale models including BERT-Tiny [10], ELECTRA-Small [11], and DistilBERT [14]. b) Base parameter scale models including BERT-Base [10], ELECTRA-Base [11], RoBERTa-Base [12], and DeBERTa-Base [13]. c) Large parameter scale models including BERT-Large [10], ELECTRA-Large [11], RoBERTa-Large [12], and DeBERTa-Large [13]. **(3)** For Co-training methods, due to scalability constraints, we only explore the effectiveness of this pipeline on combinations of BERT-Tiny with GCN and GraphSAGE. **(4)** For the topological pre-training of LMs, we conduct experiments on various datasets and different PLM basic models. In addition, we iteratively train the proposed three pre-training tasks at batch level (named TMDC) in a multi-task learning framework. Please refer to Appendix A.2 for more details.

**Implementation details.** GNNs are mainly implemented based on the DGL toolkit [49]. PLMs are obtained from Huggingface [50] and trained under a unified framework. Considering the recent rise of parameter-efficient fine-tuning, we only fine-tune the last four encoder layers of large-scale language models. Implementation details and hyperparameter selections are provided in Appendix C.

**Evaluations metric.** We investigate the performance of different baselines through two tasks: node classification and link prediction. For the node classification task, we use Accuracy and F1-Score to evaluate the model performance. For the link prediction task, we use MRR, Hits@10, Hits@50, and Hits@100 as metrics. Due to space limitations, we present the results of some node classification experiments in the main paper, and the remaining node-level experiments with link prediction results are presented in Appendix D.1 and D.2. In addition to the aforementioned datasets, we also conduct experiments on other types of datasets, as described in Appendix D.6 (a large-scale ogbn-papers100M dataset) and Appendix D.7 (two social network datasets), where readers can find detailed information on these datasets and the corresponding experimental results.

Table 3: Node classification of the three learning paradigms on six datasets. Sports corresponds to the F1 score as an indicator of experimental results and Accuracy for the rest of the data. We bold the best results for each dataset.

| Way | PLM-Based | | GNN-Based | | | | Co-Training Based | |
|---|---|---|---|---|---|---|---|---|
| | Tiny | Base | T-GCN | B-GCN | T-SAGE | B-SAGE | GCN(T) | SAGE(T) |
| Arxiv | 70.83 | 72.96 | 72.03 | 73.30 | 72.35 | **74.14** | 69.22↓ | 73.57↑ |
| Children | 49.85 | **59.91** | 57.07 | 58.11 | 57.57 | 58.74 | 54.75↓ | 59.70↑ |
| History | 83.06 | **86.09** | 84.52 | 85.04 | 84.79 | 85.12 | 83.52↓ | 85.09↑ |
| Photo | 73.75 | 77.53 | 82.42 | 82.70 | 83.25 | 83.27 | 83.32↑ | **86.64**↑ |
| Computers | 58.32 | 60.40 | 87.43 | 87.86 | 87.90 | **88.30** | 83.93↓ | 86.04↓ |
| Sports | 81.47 | 86.02 | 84.93 | 86.16 | 87.06 | **87.34** | 85.06↑ | 85.87↓ |

## 4.1 Impact of Static Modeling of Attributes on GNNs

In this subsection, we analyze the impact of different node attribute modeling methods for downstream GNNs. Table 2 represents the effect of node classification on ogbn-arxiv-TA for different GNNs given different PLMs' features. The results on other datasets can be found in Appendix D.1. Observing Table 2, we find that RevGAT performs the best among all initial node features, while GAT and SAGE exhibit the second-best performance. They are relatively less affected by the different node features, 7.54%, 7.16%, and 7.06% respectively. JKNet, APPNP, MoNet, and MLP are more influenced by the initial node features, which all come above 20. On the other hand, the node features encoded by RoBERTa, BERT, and DistilBERT generally perform better on all types of baselines. DeBERTa, which performs better on many downstream NLP-related tasks, is less effective. This may be because DeBERTa sees less corpus during pre-training, resulting in their inability to understand the semantics well when modeling the text directly on downstream tasks. Furthermore, we compare traditional shallow text encoders (e.g., Skip-Gram) with PLMs in Appendix D.4 for a more comprehensive analysis of the impact of text modeling on downstream GNNs.

Recently, LLMs (Large-Language Models) are continuing to energize areas such as knowledge graphs [51] and recommender systems [52]. It is still an open question on how to successfully apply LLMs to text-attributed graph learning. We have conducted a preliminary exploration of how to use LLMs to advance the representation learning on TAGs. Please refer to Appendix D.8 for the detailed results and discussions.

## 4.2 Pitfalls of Co-Training Paradigm

In this subsection, we analyze the performance of the Co-training paradigm versus the PLM-based, GNN-based paradigm in terms of node classification tasks. Tiny and Base in the PLM column represent BERT-Tiny and BERT-Base, respectively. T-GCN, T-SAGE, and B-GCN, B-SAGE represent the node features of BERT-Tiny and BERT-Base fed to downstream GCN and GraphSAGE respectively. GCN(T) and SAGE(T) then denote co-training BERT-Tiny with GCN and SAGE, respectively. We compare GCN(T), and SAGE(T) with the corresponding T-GCN, T-SAGE respectively. As shown in Table 3, SAGE(T) improves in four of the datasets compared to the T-SAGE methods, with a maximum improvement of 3.39% on the Photo dataset. However, GCN(T) performs worse than T-GCN on most datasets and is even weaker than BERT-Tiny 1.61% on ogbn-arxiv-TA. The Co-training framework requires simultaneous training of PLMs and GNNs. The memory requirement and time cost of this paradigm are significantly increased. In order to facilitate the co-training of PLMs and GNNs, we reduce either the batch size or the number of neighbors. The limited scalability leads to a significant reduction in the number of neighbors for GNN aggregation, which may compromise the effectiveness of message passing. To analyze the impact of scalability on the Co-Training method, we analyze the effect of the number of neighbors sampled per GNN layer on Co-training in Fig 4. As can be seen from it, the effect of the model mainly tends to increase as the number of neighbors increases. The detailed discussions on efficiency and scalability can be found in Appendix D.3.

## 4.3 Comparing PLM-based Methods with GNN-based Methods

In this subsection, we compare the PLM-based methods and the GNN-based methods in different datasets. As shown in Table 4, the column GNNs indicates the best results on all GNNs for a given PLM node feature. On the Children and History dataset, the PLM-based methods works better than

Table 4: Node classification accuracy comparison among PLM-based, GNN-based, and topological pre-training based on four datasets. The best method for each PLM on each dataset is shown in bold.

| Scale | Model | Arxiv | | | | | | History | | | | | |
|---|---|---|---|---|---|---|---|---|---|---|---|---|---|
| | | PLM | GNNs | TMLM | TDK | TCL | TMDC | PLM | GNNs | TMLM | TDK | TCL | TMDC |
| Small | BERT-Tiny | 70.83 | **72.52** | 70.83 | 71.50 | 71.55 | 71.17 | 83.06 | 85.03 | 85.76 | 85.79 | 86.06 | **86.88** |
| | ELECTRA | 71.26 | 71.12 | 72.65 | 72.83 | 73.06 | **73.71** | 84.18 | 83.11 | 84.54 | 84.42 | 84.57 | **85.18** |
| | DistilBERT | 72.50 | 74.68 | 73.53 | 74.38 | 74.89 | **75.50** | 85.81 | 85.67 | 85.76 | 86.29 | 86.28 | **86.88** |
| Base | ELECTRA | 72.67 | 71.96 | 73.51 | 74.33 | 74.26 | **75.56** | 85.64 | 83.79 | 85.77 | 85.88 | **86.62** | 86.41 |
| | BERT | 72.96 | 74.59 | 73.97 | 74.23 | 74.87 | **76.11** | 86.09 | 85.28 | 86.24 | 86.46 | 86.80 | **86.82** |
| | RoBERTa | 73.10 | 74.82 | 74.25 | 74.57 | 75.37 | **75.97** | 85.85 | 85.69 | 86.19 | 86.32 | 86.95 | **86.96** |
| | DeBERTa | 73.82 | 68.26 | 74.26 | 75.01 | 75.15 | **75.99** | 86.16 | 82.31 | 86.00 | 86.46 | **87.01** | 86.94 |
| Large | ELECTRA | 72.42 | 72.56 | 74.76 | 73.82 | 74.17 | **75.58** | 86.13 | 83.56 | 86.39 | 86.49 | **86.82** | 86.28 |
| | BERT | 73.24 | 74.68 | 75.01 | 74.31 | 75.15 | **75.75** | 86.24 | 85.15 | 86.47 | 86.73 | 86.93 | **86.94** |
| | RoBERTa | 73.83 | 74.99 | 75.18 | 74.58 | 75.48 | **75.73** | 86.41 | 85.23 | 86.72 | 86.75 | 87.11 | **87.22** |
| | DeBERTa | 74.57 | 73.59 | 75.92 | 75.20 | 75.58 | **76.20** | 87.00 | 84.89 | 87.11 | 87.26 | 87.30 | **87.32** |

| Scale | Model | Children | | | | | | Photo | | | | | |
|---|---|---|---|---|---|---|---|---|---|---|---|---|---|
| | | PLM | GNNs | TMLM | TDK | TCL | TMDC | PLM | GNNs | TMLM | TDK | TCL | TMDC |
| Small | BERT-Tiny | 49.85 | **57.86** | 54.27 | 53.43 | 54.11 | 54.66 | 73.75 | **84.12** | 74.30 | 73.99 | 73.86 | 74.92 |
| | ELECTRA | 57.03 | 56.42 | 57.35 | 56.92 | 56.88 | **58.55** | 76.58 | **83.12** | 76.09 | 76.89 | 77.74 | 77.83 |
| | DistilBERT | 59.90 | 59.33 | 60.03 | 60.23 | 60.60 | **61.38** | 77.51 | **84.34** | 77.81 | 79.69 | 81.85 | 82.52 |
| Base | ELECTRA | 59.09 | 56.42 | 59.93 | 60.27 | 60.21 | **60.83** | 77.84 | **82.98** | 78.27 | 80.18 | 81.47 | 82.82 |
| | BERT | 59.91 | 58.74 | 60.34 | 60.43 | 60.73 | **61.43** | 77.53 | **84.46** | 78.54 | 81.04 | 82.85 | 84.09 |
| | RoBERTa | 59.80 | 59.01 | 60.19 | 60.71 | 61.47 | **61.83** | 78.11 | **84.59** | 78.33 | 81.26 | 82.47 | 83.04 |
| | DeBERTa | 60.26 | 50.72 | 60.73 | 61.39 | 61.92 | **62.20** | 78.37 | 81.44 | 79.27 | 81.34 | 83.07 | **83.80** |
| Large | ELECTRA | 58.28 | 56.59 | 60.51 | 59.31 | 59.29 | **61.31** | 77.25 | **83.00** | 79.21 | 78.44 | 79.56 | 81.32 |
| | BERT | 60.65 | 58.90 | 60.84 | 61.15 | 61.50 | **62.06** | 77.72 | **84.21** | 78.95 | 79.26 | 80.74 | 81.14 |
| | RoBERTa | 60.93 | 59.26 | 62.11 | 61.95 | 62.06 | **63.24** | 79.60 | **85.12** | 80.32 | 80.82 | 81.47 | 82.55 |
| | DeBERTa | 61.61 | 56.34 | 61.91 | **62.51** | 62.37 | 62.46 | 79.63 | 82.55 | 80.45 | 81.33 | 82.33 | **82.70** |

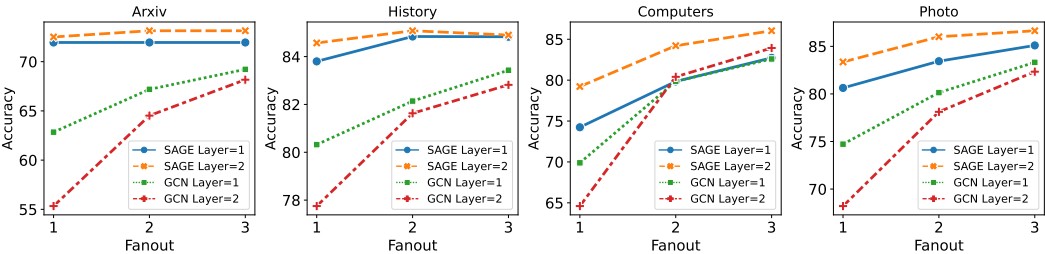

Figure 4: Node classification on four datasets are conducted to analyse the sensitivity of the two Co-Traing models, GCN(T) and SAGE(T) to the number of neighbors sampled per GNN layer. "Fanout" denotes the number of neighboring nodes to which the center node is directly connected.

the GNN-based. This is probably because their text attributes are more fully informative and thus the text attributes largely reflect the linking relationships between the nodes. Therefore the PLM-based methods model text attributes more strongly would be more advantageous in this case. And on the Photo dataset, the GNN-based method outperforms the PLM-based method across the board. This may be because the text attributes of the Photo dataset are composed of information from user reviews of the product. Some lower-quality reviews introduce a certain amount of noise to the text attributes, which will reduce the effectiveness of the PLM-based methods. In order to analyze the importance of node text attribute selection, we further conduct relevant experiments in Appendix D.9.

## 4.4 Validity of Topological Pre-training

In this subsection, we compare the three different pre-training methods with the PLM-based and GNN-based methods. We observe that on almost all PLMs and all datasets, different degrees of improvement can be achieved with the proposed pre-training methods. For these three individual pre-training tasks, we find that TCL leads to greater improvement in most cases. The difference in performance between TMLM and TCL, which both capture low-order topological structure information, indicates that learning topological structure knowledge from a node-level perspective may work better than a token-level. TDK, on the other hand, performs second only to TCL in most cases, which reflects to some extent the fact that PLMs can benefit from knowledge of the complex topology. Further, we try to combine these three pre-training tasks to optimize the model together (name TMDC). We first

perform token-level TMLM tasks on PLMs. The enhanced PLMs are then jointly optimized using both TCL and TDK tasks. (Detailed implementation can be found in Appendix A.2) Observing from the Table 4, we find that TMDC further improves performance in most cases. This indicates that different pre-training strategies can teach the PLM different topological knowledge from different perspectives, and this leads us to explore more pre-training tasks on TAGs in the future. To further analyze the effectiveness of these topological pre-training methods, we also test the performance of such topological pre-training models in other scenarios (e.g., semi-supervised learning and few-shot learning). Please refer to Appendix D.5 for detailed experimental results and discussions.

## 5 Discussion on the Practical Values

Text-attributed graphs have emerged as a prominent graph format, which finds extensive applications in modeling real-world tasks, such as the mentioned recommender systems. Our research concentrates on achieving a comprehensive understanding of the textual attributes embedded within a single node and the topological structural connections between nodes. For example, a famous example in recommender systems is the association between "diaper" and "beer", commonly co-purchased by customers, thereby establishing links between these items in the item-item graph. To achieve the optimal item representation, a prerequisite is to capture the inherent characteristics of a given item by modeling its metadata, such as title and descriptions. Simultaneously, it is imperative to incorporate valuable and unique signals derived from the graph's topological connections into the representation learning process. Given that real-world graph topology is usually shaped by human behaviors, there exist unique human perceptions and knowledge in the topology beyond the pure semantics (e.g., "diaper" and "beer" are semantically different but are connected in the co-purchased graph). Consequently, it is imperative to delve into the effective and efficient fusion of intrinsic semantics within individual nodes and the topological connections among different nodes on the text-attributed graphs. Moreover, the scope of our research also includes domains like user behavior-enhanced sponsored search, including AdsGNN [24], HBGLR [53], and PASS [54].

## 6 Conclusion

We establish the first comprehensive benchmark CS-TAG specifically designed to explore representation learning on TAGs. We collect and provide eight available text-attributed graph datasets to facilitate the NLP and GNN communities to focus and investigate the data together. Our benchmark provides a more comprehensive evaluation of different learning paradigms, validating their effectiveness and limitations. We will also continue to mine and construct more research-worthy TAGs to advance the continued healthy development of the field.

## Acknowledgement

This research was funded by the National Science Foundation of China (No. 62172443), Open Project of Xiangjiang Laboratory (22XJ03010, 22XJ03005), the Science and Technology Major Project of Changsha (No. kh2202004), Hunan Provincial Natural Science Foundation of China (No. 2022JJ30053), and the High Performance Computing Center of Central South University.

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

# A    Baselines and Model Implementation Details

## A.1    Baselines

We provide detailed descriptions on the GNNs and PLMs baselines used in the main experiments as follows.

- **GCN. [15]** Graph Convolutional Network (GCN) is a classical model that works by performing a linear approximation to spectral graph convolutions.
- **GraphSAGE. [7]** GraphSAGE is a GNN model that focuses on inductive node classification, but can also be applied for transductive settings.
- **GAT. [16]** Graph Attention Network (GAT) introduces the attention mechanism to capture the importance of neighboring nodes when aggregating information from the graph.
- **RevGAT. [33]** RevGAT combines reversible connectivity with a deep network architecture to form a deep and efficient GNN.
- **NodeFormer. [46]** NodeFormer is a scalable graph Transformer for large-scale graphs, which achieves all-pair message passing with linear complexity. In the table, we denote NodeFormer by NFormer.
- **GIN. [17]** Graph Isomorphism Network (GIN) overcomes the drawbacks of previous MPNN-based methods, which struggles to differentiate various graph structures based on the generated graph embeddings.
- **JKNet. [18]** Jumping knowledge Network (JKNet) adaptively varies the neighborhood ranges for individual node, enabling enhanced structure-aware node representation.
- **MoNet. [47]** Mixture Model Network (MoNet) captures and represents the structural properties of graphs by incorporating multiple localized perspectives of each node's neighborhood.
- **APPNP. [48]** Approximate Personalized Propagation of Neural Predictions (APPNP) is specifically developed for semi-supervised learning tasks on graph-structured data. It utilizes personalized propagation to iteratively enhance node predictions by incorporating comprehensive information from both local and global contexts.

- **DistilBERT. [14]** DistilBERT is another distilled version of BERT. The student model DistillBert shares a similar overall architecture with the teacher model BERT. DistillBert differs from BERT by removing the token-type embeddings and pooler components, and also reducing the number of layers.
- **ELECTRA. [11]** ELECTRA is a pre-training model for NLP tasks that introduces Discriminative Pre-training. ELECTRA enhances the efficiency and effectiveness by generating more efficient representations through a discriminative approach.
- **BERT. [10]** Bidirectional encoder representations from Transformers (BERT) utilizes a transformer architecture that employs self-attention mechanisms to capture word relationships within sentences. This enables the model to effectively consider both the preceding and succeeding contexts of a word, facilitating bidirectional language comprehension. BERT undergoes unsupervised pre-training on an extensive text corpus, where it predicts masked words within sentences and acquires the ability to encode contextual information.
- **RoBERTa. [12]** Robustly Optimized BERT (RoBERTa) is a variant of the BERT. RoBERTa incorporates additional modifications during pre-training to optimize its performance. In the pre-training phase, it trains on a vast corpus of unlabeled text data by employing masked language modeling (MLM) while excluding the next sentence prediction (NSP) task. RoBERTa significantly expands the amount of training data used, enabling the acquisition of more comprehensive and robust language representations.
- **DeBERTa. [13]** Decoding-enhanced BERT with Disentangled Attention (DeBERTa) brings forth two significant enhancements: disentangled attention and heterogeneous layer integration. By dividing the attention heads into distinct groups, disentangled attention empowers the model to better capture dependencies among words, enabling more targeted and specialized attention. This enhancement greatly improves the model's ability to capture long-range dependencies in a more effective manner.

In our work, we categorize three types of models, namely BERT-Tiny (4.4M), ELECTRA-Small (13.5M), and DistilBERT (66.4M), as small scale PLMs. ELECTRA-Base (109M), BERT-Base (110M), RoBERTa-Base (125M), and DeBERTa-Base (139M) are classified as base scale PLMs. ELECTRA-Large (334M), BERT-Large (340M), RoBERTa-Large (355M), and DeBERTa-Large (405M) are categorized as large scale PLMs.

## A.2 Topological Pre-training Implementation

We introduce the implementation details on 4 topological pre-training tasks, TLM, TCL, TDK and TMDC on text-attributed graphs.

**TMLM.** We adopt the Masked Language Model (MLM) training implementation in huggingface [] for the TMLM task. For a given TAG, we first pre-sample each node with five neighbors, and subsequently concatenate the text of the original node with the text of the sampled neighboring nodes. Five corresponding center-neighbor pairs are obtained for each node. We then disorganize the dataset to form a new topologically augmented dataset. The new dataset is passed into the MLM training code to pre-train the language model. The learning rate of the TMLM is set to 5e-05 on all datasets and language models. For detailed code and implementation, please refer to the project repository.

**TCL.** Traditional Contrastive Learning (CL) focuses on the central node itself (using the augmented nodes to construct the positive pairs) [42]. For our Topological Contrastive Learning (TCL) task, we consider the central node and its neighbors to form the positive pairs. In particular, when loading the data in batches, we sample one of its neighbor nodes for each central node through the adjacency matrix of the TAG. For the central node and its sampled neighbor, we compute the contrasting loss by mapping the textual representations into the contrastive space through a projection head with shared parameters by following [41]. Therefore the $\mathcal{L}_{\text{tcl}}$ in the main text can be rewritten as

$$\mathcal{L}_{\text{tcl}} = -\log \frac{\exp(\text{sim}(\boldsymbol{z}^c, \boldsymbol{z}^n)/\tau)}{\sum_{n'=1, n' \neq n}^{N} \exp(\text{sim}(\boldsymbol{z}^c, \boldsymbol{z}^{n'})/\tau)}, \tag{7}$$

where $\boldsymbol{z}^c$ is the representation of the node $c$ in the contrastive space. The main learning parameters include the learning rate $lr$ and epoch $e$. For all datasets and all language models we set $e = 5, lr = 5e - 05$. The projection head is a two-layer MLP with a hidden layer set to 128. The $\tau$ in the contrastive loss is set to 0.2. Detailed pre-training commands on each dataset can be found in the CS-TAG GitHub repository.

**TDK.** For the TDK task, we first run the Deepwalk [45] algorithm over the TAG to obtain its topological-level representation (only the topology information of TAG is fed into Deepwalk). Then we pull the representation of the center node close to the representation learned from Deepwalk. In particular, when loading the data in batches, we load the representation learned from Deepwalk at the same time. Subsequently, we from the textual representation of the center node with its corresponding topological structure representation to form the positive pairs in contrastive learning. We follow the same contrastive learning process with TCL presented above. Therefore the $\mathcal{L}_{\text{tdk}}$ in the main text can be rewritten as

$$\mathcal{L}_{\text{tdk}} = -\log \frac{\exp(\text{sim}(\boldsymbol{z}^c, \boldsymbol{k}^c)/\tau)}{\sum_{c'=1, c' \neq c}^{N} \exp(\text{sim}(\boldsymbol{z}^c, \boldsymbol{k}^{c'})/\tau)}, \tag{8}$$

For all datasets in CS-TAG, we use a uniform code to obtain the corresponding topological-level representation. The relevant parameters are consistent with the parameter settings in TCL. Detailed pre-training commands on each dataset can be found in the CS-TAG GitHub repository.

**TMDC.** TMDC attempts to pre-train the language models jointly by combining the three pre-training tasks proposed above. As the forms of the TMLM and TCL, TDK tasks differ a lot, we use an iterative training way to perform the multi-task learning. In particular, we first obtain a token-level topologically augmented language model $PLM_{tmlm}$ by TMLM. Subsequently, we combine TCL and TDK to jointly optimize the $PLM_{tmlm}$. The loss function of the joint optimization is shown as follows

$$\mathcal{L}_{\text{tcl+tdk}} = -(\log \frac{\exp(\text{sim}(\boldsymbol{z}^c, \boldsymbol{k}^c)/\tau)}{\sum_{c'=1, c' \neq c}^{N} \exp(\text{sim}(\boldsymbol{z}^c, \boldsymbol{k}^{c'})/\tau)} + \log \frac{\exp(\text{sim}(\boldsymbol{z}^c, \boldsymbol{z}^n)/\tau)}{\sum_{n'=1, n' \neq n}^{N} \exp(\text{sim}(\boldsymbol{z}^c, \boldsymbol{z}^{n'})/\tau)}). \tag{9}$$

## A.3 Code License

The code of CS-TAG uses the MIT license. Please refer to the GitHub repository for license details.

# B  Datasets

## B.1  Dataset Format

For each dataset in CS-TAG, we provide three different files. We store the graph-type data available to dgl in the .pt format in Pytorch. For the node classification dataset, two types of features are stored, the adjacency matrix and the node labels. For the Ele-Computers and Ele-Photo datasets, we also store the year features of the nodes (the year of the comments posted by the user). For the link prediction dataset, it contains only the adjacency matrix information. We use a .txt file to store the text attributes of each dataset. The .csv file stores node-id, node-label, category, and text to provide a clearer picture of the dataset for subsequent researchers.

## B.2  Dataset Construction

The construction of the text-attributed graph datasets includes the following three steps. First, pre-processing the text attributes in the original dataset, including removing missing values, removing non-English statements, removing abnormal symbols, length truncation, etc. Second, building the graph. The linking relationship between nodes has been provided in the original data of the dataset constructed in CS-TAG, such as the information of "also view", and "also buy" of the products in the Amazon dataset (indicating the two product ids that are jointly purchased or viewed), and the citation relationship between papers in DBLP, etc. Note that when obtaining the final graph data, self-edges and isolated nodes need to be removed. Third, refining the constructed graph. For the node classification dataset, the nodes in the graph need corresponding numerical node labels. We convert the categories of nodes in the original data to numerical node labels in the graph. For some datasets that are divided in a specific form (e.g., by the year of publication of a paper), we also need to store additional information about the nodes.

## B.3  Dataset Details

CS-TAG includes 8 datasets, whose details are described as follows. The statistics of the datasets is shown in Table 5.

**ogbn-arxiv-TA** dataset is derived from ogbn-arxiv. The corresponding task is to predict the categories of the papers, which is formulated as a 40-class classification problem. The text attributes of each paper node are extracted from its title and abstract in ogbn-arxiv. Note that ogbn-arxiv solely provides node embeddings acquired through shallow text encoders like Skip-Gram. Different from them, we use different PLMs to model the node attributes to get the initial node features and delve into the performance of employing multiple PLM features in downstream GNNs.

**Books-Children/History** datasets are extracted from the Amazon-Books dataset. Books-Children consists of items with the second-level label "Children", while Books-History consists of items with the second-level label "History". The nodes in the dataset are books, and the edges mean two books are frequently co-purchased or co-viewed. The label of each dataset is the three-level label of the book. We choose the title and description of the book itself as the text attributes of the node. The task is to classify books into 24 and 12 categories, respectively.

**Ele-Computers/Photo** datasets are extracted from the Amazon-Electronics dataset. Ele-Computers consists of items with the second-level label "Computers", while Ele-Photo consists of items with the second-level label "Photo". The two datasets are extracted from the updated 2018 Amazon Computer and Amazon Photo datasets [27]. The nodes in the dataset are electronics related products, and the edge between two products means that they are frequently co-purchased or co-viewed. The label of each dataset is the three-level label of the electronics products. We adopt user reviews on the item as its text attribute. Since the item has multiple reviews, we mainly adopt the review with the highest number of votes. For some items lacking highly votes reviews, we randomly adopt a user review as the text attribute. The task on the two datasets is to classify electronics products into 10 and 12 categories, respectively.

**Sports-Fitness** dataset is extracted from the Amazon-Sports dataset. It consist of items with the second-level label "Fitness". The nodes in the dataset are the fitness-related items, and the edge between two items means that they are frequently co-purchased or co-viewed. The label of the dataset is the three-level label of the items. The task on this dataset is to classify items into 13 categories.

Table 5: Statistics of text-attributed graph dataset used in CS-TAG.

| Dataset | Nodes | Edges | Class | Split Scheme | Split Ratio | Task Type | Metric | Max length |
|---|---|---|---|---|---|---|---|---|
| **ogbn-arxiv-TA** | 169,343 | 1,166,243 | 40 | Time | 54/18/28 | Node Class. | Acc, F1 | 512 |
| **Books-Children** | 76,875 | 1,554,578 | 24 | Random | 60/20/20 | Node Class. | Acc, F1 | 256 |
| **Books-History** | 41,551 | 358,574 | 12 | Random | 60/20/20 | Node Class. | Acc | 256 |
| **Ele-Computers** | 87,229 | 721,081 | 10 | Time | 72/17/11 | Node Class. | Acc, F1 | 256 |
| **Ele-Photo** | 48,362 | 500,928 | 12 | Time | 60/20/20 | Node Class. | Acc, F1 | 512 |
| **Sports-Fitness** | 173,055 | 1,773,500 | 13 | Random | 20/10/70 | Node Class. | F1 | 64 |
| **CitationV8** | 1,106,759 | 6,120,897 | - | Time | 99/1/1 | Link Prediction | MRR | 256 |
| **GoodReads** | 676,084 | 8,582,306 | - | Random | 90/2/8 | Link Prediction | Hits@K | 24 |

**CitationV8** is a directed graph dataset, representing the citation relationship among a subset of papers extracted from DBLP [38]. It is constructed following the form of ogbl-citation2 [34]. The corresponding task is to predict the missing citations given some existing citations. Specifically, for each source paper, two of its references are randomly dropped, and the prediction model tries to rank the missing two references higher than 2,000 negative reference candidates. The negative references are randomly-sampled from all the previously published papers that are not referenced by the source paper. We adopt the title and abstract of each paper as its node text attributes.

**GoodReads** dataset is extracted from the world's largest book review site Goodreads. Its nodes are books and edges are identified by similarity relationships between books provided in the website. The corresponding task is to predict the similar relationships between books. We expect the model to rank the true correlations over the false ones. Specifically, we rank each true correlation among a set of 5,000 randomly-sampled negative correlation.

### B.4 Datasets License

The datasets follow the MIT license. Please refer to the GOOD GitHub repository for license details.

## C Experiment Settings

### C.1 Experimental Settings of GNNs

We conduct experiments on 9 GNN models described in A.1 on 6 node classification datasets. We use the aforementioned 5 PLMs with different parameter scales to model the node attributes and form the initial node features of the graph data. Each experiment is repeated three times and the evaluation metrics are accuracy and F1-score. The parameters shared by all GNN models include epochs, model layers, hidden units, learning rate, and dropout ratio, and their values are set to 1000, {2,3}, {64,128,256},{1e-04 ~1e-02}, 0.2, respectively. Besides these hyperparameters, for GAT we freeze the number of heads to 3 and set the ratio of attention-drop out to 0 by default. For GraphSAGE model, we use mean pool to aggregate the neighbor information, and for JKNet, we use cat to aggregate the features. For APPNP, we set the teleport probability to 0.1 by default and the number of propagation steps is set to 2. For MoNet, we set the pseudo coordinate dimensions in GMMConv to 2 and 3, and we set the number of kernels in GMMConv layer to 2. Since it mostly does not converge at epoch=1000, we set its maximum epoch to 2000. For GIN, we set the number of mlp layers as 2. The eval patience of all models is set to 1. We use cross-entropy loss with the AdamW optimizer to train and optimize all the above models. GNNs are mainly derived from the implementation in the DGL library.

### C.2 Experimental Settings of PLMs

We conduct experiments on 5 PLMs with different parameters described in A.1 on 6 node classification dataset. Considering the efficiency of the language models, we conduct experiments on each dataset only once. The parameters shared by all the PLMs in A.1 include epochs $e$, label smoothing factor $lsf$, learning rate $lr$, warm-up epochs $w$, batch size, and eval patience. The label smoothing factor is used to calculate the cross-entropy loss which is set in GNN as well. The $w$ denotes the duration of the warm-up phase, and $w = 1$ means the duration of the warm-up is in one epoch. For all datasets we set $e$, $lsf$, $w$, $lr$ to 4, 0.1, 1, {5e-06 ~5e-04 }, respectively. Due to different model parameters and different dataset sizes, we list the eval patience, batch size of different models on each dataset in Table 6. For PLMs at the same scale, we use the same batch size. For large-scale

Table 6: Batch size and eval patience of the different scale PLMs.

| Datasets | Small Scale | | Base Scale | | Large Scale | |
|---|---|---|---|---|---|---|
| | Batch Size | Eval Patience | Batch Size | Eval Patience | Batch Size | Eval Patience |
| ogbn-arxiv-TA | 100 | 50000 | 60 | 50000 | 60 | 50000 |
| Books-Children | 240 | 15000 | 90 | 15000 | 90 | 15000 |
| Books-History | 240 | 8000 | 90 | 8000 | 90 | 8000 |
| Ele-Computeres | 300 | 20000 | 180 | 20000 | 180 | 20000 |
| Ele-Photo | 100 | 5000 | 60 | 5000 | 60 | 5000 |
| Sports-Fitness | 800 | 10000 | 400 | 10000 | 400 | 10000 |

models, since full parameter fine-tuning is costly, we only fine-tune the last four encoder layers and its' effect is sometimes better than full parameter tuning. The experimental setup of several topological pre-training methods has been mentioned in the previous section, and for the topological pre-trained language models, we follow the same tuning strategy as the PLM-based methods.

### C.3 Reproducibility

For all experiments, we select the best checkpoints according to the validation sets, and report the results. All the datasets and codes to reproduce the results in this paper are available at https://github.com/sktsherlock/TAG-Benchmark.

## D Additional Experiment Results

### D.1 Experimental Results for Node Classification

**GNN-based methods.** Table 3-5 lists all the experimental results for the node classification task. We first analyze the performance of different GNN-based methods on the tasks of node classification. Over the evaluation metric accuracy, one can see that RevGAT, GraphSAGE, and GAT perform the best on these datasets, which are less affected by the node features learned by different PLMs. While for the other GNN models, their performance is affected by the PLMs to different extents. GIN, which is commonly used for graph-level tasks, does not achieve better performance on node-level tasks. MoNet's performance on Books-Children and History datasets is even lower than MLP in most cases.

By investigating the impact of node features encoded by different PLMs on the downstream models, one can see that node features encoded by ELECTRA typically produce a larger gap between GNN and MLP. The failure of the ELECTRA model to generate higher quality node features may be due to its discriminative pre-training way. In contrast, the RoBERTa-Base model seems to have a better semantic understanding, with a difference between GNN and MLP of only 1.91 and 4.44 on the Books-History and Books-Children datasets. It is worth noting that the node features encoded by DeBERTa also perform poorly. However, DeBERTa performs better on NLP-related tasks, which indicates that the node features encoded by a language model cannot be judged by its performance on downstream tasks alone. The ineffectiveness of the DeBERTa to obtain better node features may be mainly due to its reduced corpus during pre-training.

Similar experimental conclusions can be drawn on the F1-score results as shown in Table 4. Note that the F1 score of downstream models is more significantly affected by different PLMs. For example, on Books-Children, the score of JKNet ranges from 19.79 to 30.94 under different PLMs.

**PLM-based methods and Topological Pre-training** Between Table 3 to 5, PLMs denote PLM-based methods, i.e., fine-tuning directly on the dataset without considering the topology. While TLM, TDK, TCL, and TMDC denote fine-tuning on models pre-trained for these four topological pre-training tasks. Observing the experimental results on these datasets, the PLM-basd method performs the worst on all models with all datasets. This indicates that it is not sensible to ignore topology and use only text attributes for representation learning on TAGs. In contrast, TMDC, a multi-task form of topology pre-training, performs the best, achieving the best on both metrics in the majority of experiments. This indicates that combining different topological structure pre-training methods can teach language models topological structure knowledge from different perspectives.

Table 7: Node classification accuracy on ogbn-arxiv-TA, Books-Children, Books-History and Ele-Photo. We bold the best results for each row.

**ogbn-arxiv-TA**

| Methods | | Small Scale | | | Base Scale | | | | Large Scale | | | |
|---|---|---|---|---|---|---|---|---|---|---|---|---|
| | | BERT-Tiny | ELECTRA | DistilBERT | ELECTRA | BERT | RoBERTa | DeBERTa | ELECTRA | BERT | RoBERTa | DeBERTa |
| LMs | PLM | 70.83 | 71.26 | 72.50 | 72.67 | 72.96 | 73.10 | 73.82 | 72.42 | 73.24 | 73.83 | 74.57 |
| GNNs | GCN | 72.03 | 68.45 | 73.39 | 70.81 | 73.30 | 73.56 | 68.15 | 70.44 | 73.25 | 73.95 | 72.57 |
| | GAT | 72.25 | 70.97 | 73.48 | 71.67 | 73.40 | 73.38 | 66.56 | 71.01 | 73.37 | 73.72 | 71.50 |
| | SAGE | 72.35 | 69.63 | 74.48 | 70.82 | 74.14 | 74.52 | 67.58 | 70.72 | 74.15 | 74.64 | 73.22 |
| | RevGAT | 72.52 | 71.12 | 74.68 | 71.96 | 74.59 | 74.82 | 68.26 | 72.56 | 74.68 | 74.99 | 73.59 |
| | NFormer | 71.91 | 69.45 | 73.56 | 70.43 | 72.80 | 73.12 | 67.11 | 70.04 | 73.12 | 73.12 | 71.88 |
| | GIN | 68.42 | 58.09 | 72.30 | 64.88 | 71.94 | 72.63 | 62.05 | 64.47 | 71.88 | 73.10 | 71.25 |
| | JKNet | 69.50 | 62.87 | 71.44 | 63.41 | 70.08 | 69.40 | 44.16 | 58.34 | 68.70 | 68.10 | 54.41 |
| | APPNP | 71.63 | 59.55 | 74.01 | 65.62 | 73.90 | 74.01 | 52.37 | 64.52 | 73.53 | 74.17 | 69.28 |
| | MoNet | 45.13 | 36.65 | 50.51 | 38.91 | 46.90 | 44.53 | 29.67 | 37.26 | 43.31 | 44.01 | 33.53 |
| | MLP | 57.22 | 36.58 | 68.11 | 48.56 | 67.35 | 69.31 | 47.07 | 47.72 | 66.85 | 69.51 | 66.28 |
| Co-Training | GCN | 69.22 | OOM | OOM | OOM | OOM | OOM | OOM | OOM | OOM | OOM | OOM |
| | SAGE | **73.57** | OOM | OOM | OOM | OOM | OOM | OOM | OOM | OOM | OOM | OOM |
| TPT | TMLM | 70.83 | 72.65 | 73.53 | 73.51 | 73.97 | 74.25 | 74.26 | 74.76 | 75.01 | 75.18 | 75.92 |
| | TDK | 71.50 | 72.83 | 74.38 | 74.33 | 74.23 | 74.57 | 75.01 | 73.82 | 74.31 | 74.58 | 75.20 |
| | TCL | 71.55 | 73.06 | 74.89 | 74.26 | 74.87 | 75.37 | 75.15 | 74.17 | 75.15 | 75.48 | 75.58 |
| | TMDC | 71.17 | **73.71** | **75.50** | **75.56** | **76.11** | **75.97** | **75.99** | **75.58** | **75.75** | **75.73** | **76.20** |

**Books-Children**

| Methods | | Small Scale | | | Base Scale | | | | Large Scale | | | |
|---|---|---|---|---|---|---|---|---|---|---|---|---|
| | | BERT-Tiny | ELECTRA | DistilBERT | ELECTRA | BERT | RoBERTa | DeBERTa | ELECTRA | BERT | RoBERTa | DeBERTa |
| LMs | PLM | 49.85 | 57.03 | 59.90 | 59.09 | 59.91 | 59.80 | 60.26 | 58.28 | 60.65 | 60.93 | 61.61 |
| GNNs | GCN | 57.07 | 54.35 | 58.19 | 55.31 | 58.11 | 58.62 | 50.72 | 54.66 | 57.70 | 57.11 | 54.89 |
| | GAT | 57.22 | 56.18 | 57.91 | 55.89 | 57.70 | 57.83 | 47.63 | 55.72 | 57.50 | 57.35 | 55.45 |
| | SAGE | 57.57 | 55.32 | 59.33 | 55.84 | 58.74 | 58.97 | 49.61 | 55.52 | 58.40 | 58.21 | 56.29 |
| | RevGAT | 57.86 | 56.42 | 59.28 | 56.42 | 58.67 | 59.01 | 49.63 | 56.59 | 58.90 | 59.26 | 56.34 |
| | NFormer | 56.89 | 55.12 | 58.03 | 55.12 | 57.42 | 57.26 | 48.89 | 54.59 | 57.10 | 56.43 | 54.48 |
| | GIN | 53.12 | 47.26 | 55.86 | 50.45 | 55.62 | 55.62 | 47.08 | 49.85 | 55.22 | 55.37 | 51.90 |
| | JKNet | 53.48 | 48.36 | 51.25 | 45.90 | 52.33 | 49.12 | 34.19 | 42.56 | 51.18 | 44.47 | 36.89 |
| | APPNP | 56.19 | 49.63 | 57.83 | 52.42 | 57.73 | 57.49 | 41.13 | 50.24 | 54.73 | 54.73 | 46.51 |
| | MoNet | 36.81 | 35.18 | 37.57 | 34.87 | 36.43 | 36.02 | 32.29 | 34.22 | 35.60 | 34.70 | 33.16 |
| | MLP | 48.34 | 40.33 | 53.18 | 43.14 | 52.55 | 54.57 | 43.55 | 43.11 | 52.43 | 52.61 | 48.55 |
| Co-Training | GCN | 54.75 | OOM | OOM | OOM | OOM | OOM | OOM | OOM | OOM | OOM | OOM |
| | SAGE | **59.70** | OOM | OOM | OOM | OOM | OOM | OOM | OOM | OOM | OOM | OOM |
| TPT | TMLM | 54.27 | 57.35 | 60.03 | 59.93 | 60.34 | 60.19 | 60.73 | 60.51 | 60.84 | 62.11 | 61.91 |
| | TDK | 53.43 | 56.92 | 60.23 | 60.27 | 60.43 | 60.71 | 61.39 | 59.31 | 61.15 | 61.95 | **62.51** |
| | TCL | 54.11 | 56.88 | 60.60 | 60.21 | 60.73 | 61.47 | 61.92 | 59.29 | 61.50 | 62.06 | 62.37 |
| | TMDC | 54.66 | **58.55** | **61.38** | **60.83** | **61.43** | **61.83** | **62.20** | **61.31** | **62.06** | **63.24** | 62.46 |

**Books-History**

| Methods | | Small Scale | | | Base Scale | | | | Large Scale | | | |
|---|---|---|---|---|---|---|---|---|---|---|---|---|
| | | BERT-Tiny | ELECTRA | DistilBERT | ELECTRA | BERT | RoBERTa | DeBERTa | ELECTRA | BERT | RoBERTa | DeBERTa |
| LMs | PLM | 83.06 | 84.18 | 85.81 | 85.64 | 86.09 | 85.85 | 86.16 | 86.13 | 86.24 | 86.41 | 87.00 |
| GNNs | GCN | 84.52 | 82.08 | 85.14 | 82.46 | 85.04 | 85.15 | 82.31 | 82.54 | 84.95 | 84.51 | 84.22 |
| | GAT | 84.21 | 82.85 | 84.68 | 82.97 | 84.49 | 84.76 | 80.71 | 83.10 | 84.36 | 84.47 | 83.78 |
| | SAGE | 84.79 | 82.12 | 85.56 | 82.53 | 85.12 | 85.47 | 82.00 | 82.45 | 85.08 | 84.92 | 84.51 |
| | RevGAT | 85.03 | 83.11 | 85.67 | 83.79 | 85.26 | 85.69 | 81.98 | 83.56 | 85.15 | 85.23 | 84.89 |
| | NFormer | 83.59 | 80.96 | 84.49 | 81.16 | 84.59 | 84.46 | 81.46 | 80.15 | 84.29 | 84.23 | 82.99 |
| | GIN | 82.62 | 73.69 | 83.60 | 76.29 | 83.19 | 84.01 | 79.81 | 76.47 | 83.29 | 83.34 | 82.89 |
| | JKNet | 82.97 | 80.25 | 84.01 | 79.88 | 83.53 | 83.31 | 69.26 | 76.77 | 83.45 | 82.17 | 77.36 |
| | APPNP | 84.31 | 78.65 | 85.49 | 79.91 | 85.28 | 85.35 | 78.16 | 79.18 | 84.97 | 84.86 | 82.68 |
| | MoNet | 71.24 | 69.67 | 70.41 | 66.17 | 71.28 | 72.72 | 59.17 | 60.48 | 70.66 | 66.26 | 60.79 |
| | MLP | 79.86 | 64.36 | 83.00 | 68.09 | 82.84 | 83.78 | 74.60 | 68.88 | 83.18 | 82.73 | 80.46 |
| Co-Training | GCN | 83.52 | OOM | OOM | OOM | OOM | OOM | OOM | OOM | OOM | OOM | OOM |
| | SAGE | 85.09 | OOM | OOM | OOM | OOM | OOM | OOM | OOM | OOM | OOM | OOM |
| TPT | TMLM | 85.76 | 84.54 | 85.76 | 85.77 | 86.24 | 86.19 | 86.00 | 86.39 | 86.47 | 86.72 | 87.11 |
| | TDK | 85.79 | 84.42 | 86.29 | 85.88 | 86.46 | 86.32 | 86.46 | 86.49 | 86.73 | 86.75 | 87.26 |
| | TCL | 86.06 | 84.57 | 86.28 | **86.62** | 86.80 | 86.95 | **87.01** | **86.82** | 86.95 | 87.11 | 87.30 |
| | TMDC | **86.88** | **85.18** | **86.88** | 86.41 | **86.82** | **86.96** | 86.94 | 86.28 | **86.94** | **87.22** | **87.32** |

**Ele-Photo**

| Methods | | Small Scale | | | Base Scale | | | | Large Scale | | | |
|---|---|---|---|---|---|---|---|---|---|---|---|---|
| | | BERT-Tiny | ELECTRA | DistilBERT | ELECTRA | BERT | RoBERTa | DeBERTa | ELECTRA | BERT | RoBERTa | DeBERTa |
| LMs | PLM | 73.75 | 76.58 | 77.51 | 77.84 | 77.53 | 78.11 | 78.37 | 77.25 | 77.72 | 79.60 | 79.63 |
| GNNs | GCN | 82.42 | 78.86 | 82.91 | 79.99 | 82.70 | 82.99 | 80.07 | 79.20 | 82.01 | 83.82 | 80.76 |
| | GAT | 83.82 | 82.83 | 83.75 | 82.82 | 83.74 | 83.99 | 79.47 | **83.00** | 83.48 | 83.97 | 82.55 |
| | SAGE | 83.25 | 80.90 | 83.50 | 81.79 | 83.27 | 83.81 | 81.44 | 81.05 | 82.77 | 84.15 | 81.88 |
| | RevGAT | 84.12 | **83.12** | **84.34** | **82.98** | **84.46** | **84.59** | 80.98 | 82.59 | **84.21** | **85.12** | 81.12 |
| | NFormer | 79.98 | 80.45 | 82.69 | 80.02 | 81.79 | 82.44 | 79.66 | 79.96 | 81.23 | 82.96 | 80.62 |
| | GIN | 76.09 | 64.89 | 77.22 | 68.56 | 76.55 | 77.76 | 69.91 | 66.98 | 75.32 | 79.37 | 70.93 |
| | JKNet | 79.68 | 75.29 | 80.41 | 76.53 | 79.72 | 79.13 | 60.13 | 74.35 | 79.18 | 78.61 | 69.57 |
| | APPNP | 79.24 | 70.77 | 81.45 | 73.56 | 80.68 | 81.82 | 67.52 | 72.89 | 79.21 | 82.02 | 76.86 |
| | MoNet | 76.24 | 66.55 | 72.84 | 68.89 | 73.17 | 73.67 | 57.48 | 66.24 | 71.08 | 73.57 | 61.87 |
| | MLP | 58.43 | 47.69 | 64.47 | 51.24 | 62.51 | 65.64 | 54.12 | 49.98 | 60.88 | 66.26 | 58.76 |
| Co-Training | GCN | 83.32 | OOM | OOM | OOM | OOM | OOM | OOM | OOM | OOM | OOM | OOM |
| | SAGE | **86.64** | OOM | OOM | OOM | OOM | OOM | OOM | OOM | OOM | OOM | OOM |
| TPT | TMLM | 74.30 | 76.09 | 77.81 | 78.27 | 78.54 | 78.33 | 79.27 | 79.21 | 78.95 | 80.32 | 80.45 |
| | TDK | 73.99 | 76.89 | 79.69 | 80.18 | 81.04 | 81.26 | 81.34 | 78.44 | 79.26 | 80.82 | 81.33 |
| | TCL | 73.86 | 77.74 | 81.85 | 81.47 | 82.85 | 82.47 | 83.07 | 79.56 | 80.74 | 81.47 | 82.33 |
| | TMDC | 74.92 | 77.83 | 82.52 | 82.82 | 84.09 | 83.04 | **83.80** | 81.32 | 81.14 | 82.55 | **82.70** |

Table 8: Node classification f1 score on ogbn-arxiv-TA, Books-Children, Ele-Photo, and Sports-Fitness. We bold the best results for each row.

| Methods | | ogbn-arxiv-TA | | | | | | | | | | |
| | | Small Scale | | | Base Scale | | | | Large Scale | | | |
| | | BERT-Tiny | ELECTRA | DistilBERT | ELECTRA | BERT | RoBERTa | DeBERTa | ELECTRA | BERT | RoBERTa | DeBERTa |
| LMs | PLM | 50.48 | 52.15 | 53.01 | 54.81 | 54.39 | 55.41 | 56.40 | 55.71 | 55.88 | 57.52 | 57.64 |
| GNNs | GCN | 51.38 | 44.40 | 54.08 | 48.91 | 53.44 | 54.32 | 32.38 | 47.17 | 53.14 | 54.35 | 50.97 |
| | GAT | 52.26 | 49.33 | 53.97 | 51.41 | 54.11 | 53.61 | 41.04 | 53.79 | 50.62 | 53.88 | 51.32 |
| | SAGE | 51.81 | 45.58 | 55.22 | 49.38 | 54.57 | 55.86 | 38.56 | 48.37 | 54.39 | 55.79 | 52.41 |
| | RevGAT | 52.34 | 49.88 | 55.12 | 52.21 | 54.68 | 55.91 | 42.06 | 54.11 | 55.21 | 56.04 | 53.01 |
| | NFormer | 49.95 | 45.56 | 51.02 | 47.75 | 51.16 | 51.58 | 39.55 | 41.12 | 50.03 | 51.16 | 47.89 |
| | GIN | 46.92 | 22.56 | 52.28 | 41.88 | 52.24 | 52.31 | 32.76 | 39.14 | 51.39 | 52.58 | 49.88 |
| | JKNet | 47.10 | 36.82 | 47.07 | 37.25 | 48.49 | 45.75 | 29.56 | 31.27 | 44.54 | 43.29 | 28.55 |
| | APPNP | 50.63 | 33.67 | 54.32 | 41.50 | 54.35 | 54.26 | 40.90 | 40.58 | 53.68 | 54.12 | 42.69 |
| | MoNet | 38.62 | 20.15 | 35.26 | 24.56 | 36.57 | 39.22 | 28.45 | 20.12 | 31.26 | 34.28 | 26.51 |
| | MLP | 34.71 | 15.35 | 47.37 | 25.33 | 45.88 | 48.05 | 24.39 | 45.58 | 23.38 | 48.37 | 43.03 |
| Co-Training | GCN | 51.55 | OOM | OOM | OOM | OOM | OOM | OOM | OOM | OOM | OOM | OOM |
| | SAGE | **52.76** | OOM | OOM | OOM | OOM | OOM | OOM | OOM | OOM | OOM | OOM |
| TPT | TMLM | 50.81 | 52.43 | 54.41 | 55.02 | 54.89 | 56.67 | 56.92 | 55.83 | 56.42 | 58.60 | 58.73 |
| | TDK | 50.96 | 52.88 | 54.96 | 55.52 | 55.21 | 57.03 | 57.23 | 56.22 | 56.96 | 59.44 | 59.59 |
| | TCL | 51.46 | 53.11 | 55.36 | 56.01 | 55.96 | 57.68 | 57.96 | 57.03 | 57.69 | 60.02 | 60.13 |
| | TMDC | 51.79 | **53.45** | **55.96** | **56.45** | **56.39** | **58.11** | **58.45** | **57.64** | **58.12** | **60.56** | **60.78** |

| Methods | | Books-Children | | | | | | | | | | |
| | | Small Scale | | | Base Scale | | | | Large Scale | | | |
| | | BERT-Tiny | ELECTRA | DistilBERT | ELECTRA | BERT | RoBERTa | DeBERTa | ELECTRA | BERT | RoBERTa | DeBERTa |
| LMs | PLM | 26.26 | 38.09 | 46.17 | 46.44 | 47.55 | 48.25 | 49.54 | 40.70 | 50.51 | 50.67 | 51.34 |
| GNNs | GCN | 51.38 | 44.40 | 54.08 | 41.90 | 46.35 | 46.99 | 33.97 | 41.57 | 46.16 | 45.89 | 40.93 |
| | GAT | 52.26 | 49.33 | 53.97 | 45.60 | 46.55 | 48.54 | 29.88 | 46.16 | 48.46 | 45.98 | 43.46 |
| | SAGE | 51.81 | 45.58 | 55.22 | 43.49 | 47.31 | 48.77 | 33.30 | 43.60 | 46.74 | 47.77 | 42.94 |
| | RevGAT | 52.35 | **49.96** | **55.52** | 45.78 | 47.56 | 49.01 | 33.41 | 46.26 | 48.59 | 47.82 | 46.29 |
| | NFormer | 50.30 | 43.32 | 51.63 | 42.23 | 44.43 | 44.12 | 28.55 | 40.06 | 44.15 | 44.26 | 40.96 |
| | GIN | 46.92 | 22.56 | 52.28 | 38.16 | 44.21 | 44.92 | 27.92 | 37.56 | 44.20 | 43.22 | 41.09 |
| | JKNet | 47.10 | 36.82 | 47.07 | 30.15 | 38.92 | 30.35 | 7.98 | 23.14 | 36.22 | 26.56 | 13.54 |
| | APPNP | 50.63 | 33.67 | 54.32 | 37.53 | 46.08 | 44.22 | 16.16 | 32.00 | 45.65 | 40.95 | 23.53 |
| | MoNet | 36.89 | 21.56 | 43.38 | 28.56 | 34.45 | 36.51 | 23.65 | 31.12 | 36.64 | 37.99 | 29.92 |
| | MLP | 34.71 | 15.35 | 47.37 | 20.75 | 34.88 | 38.87 | 22.54 | 21.14 | 34.15 | 36.85 | 29.97 |
| Co-Training | GCN | 52.03 | OOM | OOM | OOM | OOM | OOM | OOM | OOM | OOM | OOM | OOM |
| | SAGE | **52.96** | OOM | OOM | OOM | OOM | OOM | OOM | OOM | OOM | OOM | OOM |
| TPT | TMLM | 36.47 | 42.04 | 48.16 | 44.79 | 47.87 | 48.37 | 49.42 | 46.75 | 48.89 | 51.80 | 51.59 |
| | TDK | 33.22 | 40.26 | 46.80 | 47.16 | 49.40 | 49.42 | 51.84 | 45.82 | 50.06 | 52.07 | 52.21 |
| | TCL | 34.15 | 40.43 | 48.46 | 42.93 | 49.77 | 49.72 | 51.85 | 44.59 | 50.62 | **52.63** | 51.85 |
| | TMDC | 35.78 | 44.21 | 49.31 | **46.96** | **50.89** | **50.09** | **52.38** | **49.96** | **51.21** | 52.55 | **52.28** |

| Methods | | Ele-Photo | | | | | | | | | | |
| | | Small Scale | | | Base Scale | | | | Large Scale | | | |
| | | BERT-Tiny | ELECTRA | DistilBERT | ELECTRA | BERT | RoBERTa | DeBERTa | ELECTRA | BERT | RoBERTa | DeBERTa |
| LMs | PLM | 66.11 | 69.31 | 70.82 | 70.48 | 70.04 | 71.21 | 72.10 | 67.97 | 70.96 | 73.69 | 74.30 |
| GNNs | GCN | 75.03 | 69.32 | 76.44 | 72.12 | 75.38 | 76.49 | 70.22 | 70.90 | 74.91 | 76.84 | 72.93 |
| | GAT | 77.24 | 76.22 | 77.26 | 76.54 | 76.97 | 77.30 | 73.02 | 75.99 | 76.77 | 77.47 | 75.66 |
| | SAGE | 75.75 | 72.97 | 76.22 | 73.96 | 75.26 | 76.71 | 73.25 | 73.53 | 74.86 | 77.31 | 74.11 |
| | RevGAT | **77.34** | **76.55** | **77.31** | 76.81 | 77.23 | 77.35 | 73.55 | **76.12** | **76.94** | **77.51** | 76.21 |
| | NFormer | 72.26 | 68.12 | 73.22 | 69.57 | 72.15 | 73.28 | 68.12 | 73.49 | 73.56 | 74.45 | 70.12 |
| | GIN | 67.95 | 53.35 | 68.75 | 60.26 | 69.45 | 63.50 | 51.77 | 55.90 | 62.82 | 70.48 | 57.46 |
| | JKNet | 69.99 | 64.72 | 69.92 | 65.5 | 69.5 | 67.02 | 40.33 | 62.51 | 69.72 | 68.11 | 54.92 |
| | APPNP | 69.86 | 56.59 | 72.84 | 61.91 | 71.53 | 73.37 | 48.02 | 59.36 | 68.74 | 73.88 | 63.26 |
| | MoNet | 45.51 | 44.12 | 51.12 | 48.86 | 52.26 | 55.51 | 34.56 | 42.56 | 53.35 | 57.11 | 38.69 |
| | MLP | 39.60 | 14.98 | 48.32 | 24.56 | 44.86 | 50.65 | 29.30 | 23.25 | 41.91 | 51.75 | 37.18 |
| Co-Training | GCN | 74.59 | OOM | OOM | OOM | OOM | OOM | OOM | OOM | OOM | OOM | OOM |
| | SAGE | 76.98 | OOM | OOM | OOM | OOM | OOM | OOM | OOM | OOM | OOM | OOM |
| TPT | TMLM | 65.19 | 66.62 | 71.50 | 70.94 | 71.34 | 71.76 | 72.27 | 72.77 | 73.23 | 73.96 | 74.61 |
| | TDK | 66.15 | 68.08 | 72.92 | 73.60 | 74.98 | 75.08 | 75.84 | 73.79 | 71.71 | 74.85 | 75.85 |
| | TCL | 64.94 | 69.34 | 75.03 | 74.93 | 76.59 | 76.62 | 76.92 | 71.95 | 75.05 | 75.48 | **76.95** |
| | TMDC | 66.23 | 70.02 | 76.82 | **76.97** | **78.18** | **77.45** | **78.69** | 74.47 | 75.27 | 77.08 | 76.10 |

| Methods | | Sports-Fitness | | | | | | | | | | |
| | | Small Scale | | | Base Scale | | | | Large Scale | | | |
| | | BERT-Tiny | ELECTRA | DistilBERT | ELECTRA | BERT | RoBERTa | DeBERTa | ELECTRA | BERT | RoBERTa | DeBERTa |
| LMs | PLM | 81.47 | 76.58 | 85.31 | 83.94 | 86.02 | 83.04 | 84.28 | 76.85 | 85.06 | 84.16 | 86.21 |
| GNNs | GCN | 84.93 | 82.24 | 86.15 | 83.46 | 86.16 | 85.83 | 77.96 | 83.54 | 85.70 | 86.07 | 83.71 |
| | GAT | 86.45 | 85.46 | 86.58 | 85.57 | 86.65 | 86.26 | 74.23 | 86.85 | 85.80 | 85.77 | 84.56 |
| | SAGE | 87.06 | 85.06 | 87.51 | 85.70 | 87.34 | 87.39 | 76.86 | 85.86 | 87.46 | 87.58 | 85.60 |
| | RevGAT | **87.55** | 85.96 | 87.88 | 85.89 | 87.46 | 87.56 | 77.79 | **87.02** | 87.68 | **87.96** | 86.62 |
| | NFormer | 83.69 | 81.15 | 84.56 | 82.20 | 84.42 | 84.66 | 74.12 | 82.26 | 84.49 | 85.02 | 82.26 |
| | GIN | 81.59 | 71.31 | 74.91 | 71.51 | 81.95 | 72.47 | 67.06 | 71.80 | 68.77 | 83.83 | 79.86 |
| | JKNet | 80.70 | 73.56 | 75.69 | 71.70 | 80.31 | 74.37 | 15.38 | 66.20 | 80.96 | 77.24 | 48.73 |
| | APPNP | 83.62 | 71.84 | 83.54 | 73.10 | 84.59 | 82.61 | 36.81 | 73.37 | 85.18 | 84.31 | 77.37 |
| | MoNet | 59.95 | 45.56 | 68.89 | 51.23 | 69.02 | 69.56 | 55.56 | 58.59 | 70.01 | 72.12 | 65.13 |
| | MLP | 68.47 | 43.66 | 76.09 | 49.56 | 73.69 | 74.90 | 52.86 | 56.33 | 74.48 | 77.37 | 71.41 |
| Co-Training | GCN | 85.06 | 83.21 | OOM | OOM | OOM | OOM | OOM | OOM | OOM | OOM | OOM |
| | SAGE | 85.87 | **86.21** | OOM | OOM | OOM | OOM | OOM | OOM | OOM | OOM | OOM |
| TPT | TMLM | 81.56 | 80.15 | 86.22 | 84.45 | 86.35 | 83.95 | 85.16 | 79.89 | 85.45 | 84.85 | 86.51 |
| | TDK | 81.40 | 82.33 | 86.48 | 85.51 | 86.88 | 84.92 | 86.04 | 80.47 | 85.65 | 85.50 | 86.75 |
| | TCL | 82.04 | 83.20 | 87.44 | 86.72 | 88.08 | 86.67 | 87.29 | 82.05 | 86.64 | 85.80 | 87.57 |
| | TMDC | 83.15 | 84.21 | **88.56** | **87.69** | **89.01** | **88.89** | **88.56** | 84.51 | **87.89** | 86.43 | **88.78** |

Table 9: Node classification accuracy and f1 score on Ele-Computers. We bold the best results for each row.

| Methods | | Ele-Computers (Accuracy) | | | | | | | | | | |
| | | Small Scale | | | Base Scale | | | | Large Scale | | | |
| | | BERT-Tiny | ELECTRA | DistilBERT | ELECTRA | BERT | RoBERTa | DeBERTa | ELECTRA | BERT | RoBERTa | DeBERTa |
| LMs | PLM | 58.32 | 59.88 | 60.51 | 60.80 | 60.40 | 61.10 | 61.68 | 59.63 | 60.70 | 61.22 | 61.96 |
| GNNs | GCN | 87.43 | 84.13 | 88.37 | 86.10 | 87.86 | 88.50 | 82.30 | 84.79 | 87.38 | 88.56 | 85.75 |
| | GAT | 88.57 | 87.86 | 89.07 | 88.14 | 88.63 | 88.80 | 83.42 | 88.07 | 88.64 | 88.88 | 87.40 |
| | SAGE | 87.90 | 86.43 | 88.67 | 86.97 | 88.30 | 88.87 | 82.81 | 87.16 | 88.35 | 88.66 | 86.85 |
| | RevGAT | **88.66** | **87.92** | **89.32** | **88.22** | **88.72** | **88.91** | **84.12** | **88.45** | **88.86** | **89.03** | **87.56** |
| | NFormer | 86.81 | 85.43 | 87.96 | 86.05 | 86.95 | 87.12 | 81.56 | 86.26 | 87.23 | 87.65 | 84.46 |
| | GIN | 78.07 | 83.18 | 83.18 | 73.36 | 83.02 | 83.98 | 71.53 | 71.64 | 81.20 | 84.09 | 75.13 |
| | JKNet | 85.05 | 85.31 | 85.31 | 79.37 | 85.08 | 83.91 | 55.15 | 73.67 | 83.32 | 83.66 | 70.39 |
| | APPNP | 83.62 | 84.96 | 84.96 | 78.15 | 84.51 | 84.97 | 62.09 | 76.51 | 83.83 | 74.90 |
| | MoNet | 78.30 | 77.76 | 77.76 | 62.08 | 77.67 | 74.53 | 47.68 | 60.24 | 68.18 | 70.63 | 59.67 |
| | MLP | 46.02 | 53.76 | 53.76 | 36.21 | 50.85 | 55.11 | 42.45 | 36.25 | 47.59 | 55.03 | 46.74 |
| Co-Training | GCN | 83.93 | OOM | OOM | OOM | OOM | OOM | OOM | OOM | OOM | OOM | OOM |
| | SAGE | 86.04 | OOM | OOM | OOM | OOM | OOM | OOM | OOM | OOM | OOM | OOM |
| TPT | TMLM | 58.32 | 60.20 | 61.42 | 61.53 | 63.43 | 61.30 | 61.74 | 61.99 | 62.41 | 63.12 | 63.43 |
| | TDK | 58.18 | 60.42 | 63.38 | 63.13 | 62.95 | 63.80 | 64.97 | 60.96 | 62.49 | 62.92 | 62.95 |
| | TCL | 58.80 | 61.04 | 66.50 | 67.02 | 64.93 | 67.94 | 70.08 | 61.50 | 63.34 | 64.55 | 64.93 |
| | TMDC | 58.86 | 61.17 | 66.94 | 68.24 | 69.45 | 67.50 | 69.71 | 65.02 | 65.74 | 65.79 | 66.08 |

| Methods | | Ele-Computers (F1 score) | | | | | | | | | | |
| | | Small Scale | | | Base Scale | | | | Large Scale | | | |
| | | BERT-Tiny | ELECTRA | DistilBERT | ELECTRA | BERT | RoBERTa | DeBERTa | ELECTRA | BERT | RoBERTa | DeBERTa |
| LMs | PLM | 44.55 | 47.65 | 51.47 | 48.77 | 52.53 | 53.39 | 52.10 | 48.46 | 51.57 | 55.28 | 55.04 |
| GNNs | GCN | 80.88 | 75.73 | 82.24 | 79.11 | 81.03 | 82.13 | 72.97 | 78.10 | 80.93 | 82.85 | 76.56 |
| | GAT | 83.49 | 83.52 | 84.14 | 83.35 | 83.99 | 83.14 | 78.30 | 83.33 | 83.55 | 83.43 | 83.06 |
| | SAGE | 82.85 | 81.38 | 84.21 | 82.18 | 83.67 | 84.44 | 76.42 | 81.81 | 82.93 | 83.13 | 80.98 |
| | RevGAT | 83.65 | **83.76** | **84.55** | **83.56** | 84.43 | **84.69** | **79.02** | 83.46 | **83.69** | **87.79** | **83.51** |
| | NFormer | 80.57 | 75.12 | 80.26 | 77.56 | 80.01 | 81.16 | 71.03 | 76.56 | 79.65 | 81.51 | 75.55 |
| | GIN | 71.56 | 57.99 | 76.46 | 67.39 | 77.00 | 78.09 | 61.03 | 64.63 | 75.43 | 77.57 | 62.25 |
| | JKNet | 77.01 | 67.90 | 73.99 | 67.83 | 75.02 | 70.26 | 41.49 | 63.58 | 72.35 | 69.78 | 58.30 |
| | APPNP | 75.49 | 62.04 | 78.02 | 69.04 | 76.99 | 73.46 | 42.05 | 68.22 | 77.79 | 72.96 | 59.96 |
| | MoNet | 48.56 | 35.22 | 52.69 | 40.03 | 49.86 | 53.60 | 43.65 | 39.88 | 45.65 | 54.12 | 48.55 |
| | MLP | 32.20 | 16.23 | 40.06 | 21.00 | 37.41 | 41.22 | 26.21 | 19.88 | 34.25 | 42.34 | 32.73 |
| Co-Training | GCN | 80.79 | OOM | OOM | OOM | OOM | OOM | OOM | OOM | OOM | OOM | OOM |
| | SAGE | **83.67** | OOM | OOM | OOM | OOM | OOM | OOM | OOM | OOM | OOM | OOM |
| TPT | TMLM | 44.91 | 48.01 | 51.51 | 52.01 | 52.51 | 53.00 | 53.86 | 55.47 | 55.71 | 55.81 | 57.56 |
| | TDK | 44.50 | 48.48 | 52.97 | 55.80 | 55.15 | 56.53 | 59.39 | 50.44 | 55.53 | 55.94 | 54.05 |
| | TCL | 44.46 | 48.81 | 54.66 | 55.44 | 59.25 | 60.58 | 57.69 | 50.22 | 54.48 | 57.86 | 55.67 |
| | TMDC | 45.05 | 49.19 | 54.91 | 55.93 | 59.12 | 61.35 | 59.39 | 58.10 | 58.44 | 60.52 | 60.06 |

## D.2 Experimental Results for Link Prediction

We further show the experiment results for link prediction in Table 10. We mainly follow the code used for link prediction in OGB. We choose GCN and GraphSAGE as the backbone model in the link prediction experiments.

Table 10: Link Prediction results on CitationV8 and GoodReads.

| | Methods | CitationV8 | | | | GoodReads | | | |
| | | Hits10 | Hits50 | Hits100 | MRR | Hits10 | Hits50 | Hits100 | MRR |
| PLM | BERT-Tiny | 33.56 ± 1.56 | 48.15 ± 2.02 | 66.56 ± 0.56 | 41.23 ± 1.39 | 36.86 ± 2.04 | 52.45 ± 1.69 | 76.23 ± 1.11 | 42.15 ± 0.86 |
| | BERT-Base | 38.86 ± 2.53 | 57.53 ± 1.96 | 72.44 ± 0.98 | 44.56 ± 1.23 | 43.96 ± 2.26 | 60.87 ± 1.43 | 79.22 ± 0.46 | 44.43 ± 1.15 |
| GNN | T-GCN | 50.89 ± 3.56 | 74.26 ± 2.16 | 90.23 ± 0.89 | 60.79 ± 0.28 | 61.47 ± 4.65 | 84.14 ± 2.15 | 90.43 ± 0.60 | 69.44 ± 0.56 |
| | T-SAGE | 45.12 ± 3.26 | 66.23 ± 1.56 | 89.36 ± 0.99 | 54.64 ± 1.03 | 64.52 ± 3.18 | 82.65 ± 1.45 | 88.53 ± 0.61 | 74.36 ± 0.84 |
| | B-GCN | 50.39 ± 4.56 | 75.12 ± 2.56 | 90.16 ± 0.46 | 60.04 ± 0.86 | 55.42 ± 5.83 | 85.04 ± 2.34 | 91.49 ± 1.25 | 65.12 ± 0.48 |
| | B-SAGE | 44.12 ± 4.12 | 71.26 ± 1.67 | 89.12 ± 0.75 | 53.96 ± 1.24 | 54.05 ± 3.03 | 82.87 ± 0.89 | 89.61 ± 0.33 | 65.68 ± 1.13 |
| TCL | BERT-Tiny | 41.26 ± 1.49 | 57.26 ± 1.59 | 72.62 ± 0.96 | 47.26 ± 1.23 | 45.47 ± 2.53 | 61.56 ± 1.56 | 82.43 ± 0.49 | 55.12 ± 0.84 |
| | BERT-Base | 46.58 ± 1.69 | 65.77 ± 2.01 | 72.46 ± 0.56 | 52.78 ± 2.03 | 52.59 ± 2.27 | 65.97 ± 1.23 | 85.56 ± 0.34 | 61.21 ± 1.23 |
| | T-GCN | 65.23 ± 2.36 | 81.23 ± 1.36 | 92.56 ± 0.56 | 65.69 ± 0.42 | 69.58 ± 2.29 | 88.89 ± 0.56 | 93.12 ± 0.57 | 83.16 ± 1.04 |
| | T-SAGE | 61.89 ± 3.21 | 80.12 ± 1.46 | 90.23 ± 0.89 | 55.70 ± 0.15 | 70.28 ± 2.36 | 85.12 ± 0.47 | 90.38 ± 0.57 | 81.12 ± 1.23 |
| | B-GCN | **68.26 ± 2.66** | **84.56 ± 0.56** | **93.68 ± 0.26** | **70.16 ± 0.38** | **73.87 ± 3.36** | **92.82 ± 0.16** | **95.85 ± 0.15** | **85.12 ± 1.56** |
| | B-SAGE | 62.36 ± 3.46 | 80.56 ± 1.56 | 92.56 ± 0.68 | 60.18 ± 0.15 | 75.16 ± 2.26 | 90.74 ± 0.16 | 94.01 ± 0.11 | 82.15 ± 1.15 |

On the CitationV8 dataset, the parameters shared by the GCN and GraphSAGE include epochs, model layers, hidden units, learning rate, and batch size, and their values are set to 200, {2, 3}, {128, 256},{1e-04 ~1e-03}, 65536. We validate the model performance with the Hits@K and MRR metrics. We bold the best results on each row. In Table 10, the PLM indicates that we use LMs directly for link prediction. In GNN, we use the features generated by BERT-Tiny and BERT-Base to feed into GCN and SAGE, respectively. In TCL, we are subdivided into two approaches. The first is to pre-train LMs with TCL tasks and then finetune LMs for downstream link prediction tasks. The second is to use LMs pre-trained by TCL to encode text attributes. T-GCN(SAGE) denotes the node features obtained by BERT-Tiny. B-GCN(SAGE) denotes the node features obtained by BERT-Base. When we use TCL to pre-train the language models, we find that BERT-Tiny and BERT-Base improve on MRR by 6.03 and 8.22, respectively. We also find that the features obtained using TCL-BERT-Base

can significantly improve the results on downstream GNNs. For example, B-GCN improves by 10% on average on all evaluation criteria.

On the GoodReads dataset, the parameters shared by the GCN and GraphSAGE include epochs, model layers, hidden units, learning rate, and batch size, and their values are set to 200, {2}, {128},{1e-04 ~5e-03}, 524288. We validate the model performance with the Hits@K and MRR metrics. We bold the best results on each row. Observing from Table 10 we can find that after using TCL to pre-train the language models and obtaining the corresponding node features, the significant improvements are obtained in all experiments. In addition to the improvement in performance, the variance of the model is also reduced in most experiments. This indicates that the node features generated by the language model trained by a suitable pre-training task can not only enhance the performance of the downstream model but also improve the stability of the downstream model to some extent.

### D.3 Efficiency and Scalability of Co-Training Paradigm

Table 11: Effectiveness and Scalability of Co-Training with TCL.

| Datasets | BERT-Tiny | | | | | | BERT-Base | | | | | |
| | Co-Training | | | TCL | | | Co-Training | | | TCL | | |
| | Acc | Memory | Time | Acc | Memory | Time | Acc | Memory | Time | Acc | Memory | Time |
|---|---|---|---|---|---|---|---|---|---|---|---|---|
| Arxiv | 73.57 | 76.27% | 44.0 | 71.55 | 27.59% | 7.0 | - | OOM | - | **74.87** | 70.73% | 130 |
| Children | 59.70 | 97.28% | 15.5 | 54.11 | 19.76% | 2.0 | - | OOM | - | **60.73** | 80.99% | 30 |
| History | 85.09 | 85.74% | 5.7 | 86.06 | 14.69% | 1.3 | - | OOM | - | **86.80** | 98.73% | 18 |
| Photo | **86.64** | 97.83% | 14.6 | 73.86 | 22.75% | 3.1 | - | OOM | - | 82.85 | 70.65% | 120 |
| # Average | 76.25 | 89.38% | 19.95 | 71.40 | 21.20% | 3.4 | - | OOM | - | **76.31** | 80.28% | 74.50 |

In this subsection, we present the accuracy, GPU memory cost on a single 32GB V100, and the total training time (min) of the co-training paradigm and topological PLMs across five datasets. BERT-Tiny+SAGE and BERT-Base+SAGE are selected as the co-training approaches. As one of the best topological pre-training tasks, topological contrastive learning (TCL) is selected to enhance the PLMs. The experimental results are presented in the Table 11. For the small language model BERT-Tiny, the co-training paradigm costs more memories ($\sim 4X$) compared to the TCL and is much slower ($\sim 6X$) than TCL. If we use the larger language model like BERT-Base, co-training models will be out-of-memory in a single 32GB V100. Thus, the proposed topological pre-training paradigm is more efficient and practical than the co-training ones.

### D.4 Comparison between Shallow Text Encoders and PLMs

The traditional GNN pipelines generally encode the textual attributes of each node using a shallow model such as the Skip-Gram. Therefore, we extend our node text encoding approach by incorporating shallow models like Skip-Gram [29] and GloVe [30]. Experimental results (accuracy) on node classification task over ogbn-arxiv-TA and Books-Children datasets are presented in the Table 12. One can clearly see that GNNs equipped with deeper text encoders consistently outperform those with shallow encoders, verifying the importance of node attribute understanding.

Table 12: Node classification performance on shallow text encoders and PLMs. We bold the best result for each row.

| Text Attribute Encoder | | ogbn-arxiv-TA | | | | | Books-Children | | | | |
| | | GCN | GAT | SAGE | RevGAT | NFormer | GCN | GAT | SAGE | RevGAT | NFormer |
|---|---|---|---|---|---|---|---|---|---|---|---|
| Shallow | Skip-Gram | 71.97 | 72.22 | 72.02 | 73.42 | 71.03 | 56.23 | 55.84 | 56.49 | 57.13 | 56.03 |
| | GloVe | 72.12 | 72.54 | 72.48 | 73.51 | 72.04 | 57.02 | 56.58 | 57.22 | 58.12 | 56.86 |
| PLMs | DistilBERT | 73.39 | **73.48** | 74.48 | 74.68 | **73.56** | 58.19 | **57.91** | **59.33** | **59.28** | **58.03** |
| | BERT-Base | 73.30 | 73.40 | 74.14 | 74.59 | 72.80 | 58.11 | 57.70 | 58.74 | 58.67 | 57.42 |
| | RoBERTa-Base | **73.56** | 73.38 | **74.52** | **74.82** | 73.12 | **58.62** | 57.83 | 58.97 | 59.01 | 57.26 |

### D.5 Topological Pre-training under Semi-supervised and Few-shot Learning Scenarios

In order to provide the more comprehensive evaluation and deeper insights into our topological pre-training strategies, we evaluate them in different scenarios.

### D.5.1 Semi-supervised Learning

To effectively evaluate the performance of various methods within semi-supervised settings, we've adjusted the training ratio from 20% to 100%. 20% implies that only 20% of training samples are used in the model training process. BERT-Base is selected as the foundational text encoder model. Detailed experimental results concerning the node classification task across four datasets have been presented in Table 13. Notably, as the training ratio decreases, the benefits of topological pre-training approaches become even more significant. Among these strategies, TMDC demonstrates superior performance across all datasets within the semi-supervised context.

Table 13: Topological pre-training methods under semi-supervised scenarios with different training ratios

| Methods | Arxiv | | | | | History | | | | |
|---|---|---|---|---|---|---|---|---|---|---|
| | 20% | 40% | 60% | 80% | 100% | 20% | 40% | 60% | 80% | 100% |
| PLM | 57.76 | 62.56 | 67.12 | 70.15 | 72.96 | 70.86 | 75.61 | 80.18 | 83.28 | 86.09 |
| TMLM | 62.15 | 65.51 | 69.01 | 71.76 | 73.97 | 74.39 | 77.70 | 81.21 | 84.03 | 86.24 |
| TDK | 63.86 | 66.78 | 70.01 | 71.95 | 74.23 | 75.46 | 78.93 | 82.17 | 84.18 | 86.46 |
| TCL | 64.98 | 67.89 | 71.22 | 73.56 | 74.87 | 77.12 | 80.53 | 83.46 | 85.49 | 86.80 |
| TMDC | **65.48** | **68.91** | **72.92** | **75.09** | **76.11** | **78.24** | **81.48** | **84.23** | **85.80** | **86.82** |
| Methods | Children | | | | | Photo | | | | |
| | 20% | 40% | 60% | 80% | 100% | 20% | 40% | 60% | 80% | 100% |
| PLM | 44.21 | 49.12 | 53.80 | 56.97 | 59.91 | 56.61 | 63.28 | 69.45 | 73.86 | 77.53 |
| TMLM | 48.52 | 51.88 | 55.38 | 58.13 | 60.34 | 66.72 | 70.08 | 73.58 | 76.33 | 78.54 |
| TDK | 49.46 | 52.98 | 56.21 | 58.15 | 60.43 | 70.07 | 73.59 | 76.82 | 78.76 | 81.04 |
| TCL | 51.21 | 54.67 | 57.59 | 59.55 | 60.73 | 73.20 | 76.66 | 79.58 | 81.54 | 82.85 |
| TMDC | **52.88** | **56.17** | **58.91** | **60.41** | **61.43** | **75.54** | **78.83** | **81.57** | **83.07** | **84.09** |

### D.5.2 Few-shot Learning

Table 14: Topological Pre-training methods under few-shot scenarios

| Methods | Arxiv | | Children | | History | | Sports | |
|---|---|---|---|---|---|---|---|---|
| | 3-shot | 5-shot | 3-shot | 5-shot | 3-shot | 5-shot | 3-shot | 5-shot |
| PLM | 37.76 | 41.56 | 26.55 | 30.15 | 32.52 | 37.78 | 42.56 | 46.22 |
| TMLM | 40.08 | 45.21 | 31.86 | 36.89 | 35.51 | 40.15 | 44.18 | 50.15 |
| TDK | 41.15 | 47.42 | 34.56 | 39.26 | 36.58 | 42.26 | 45.69 | 52.68 |
| TCL | 43.26 | 49.58 | 38.26 | 42.12 | 38.95 | 44.12 | 48.58 | 55.64 |
| TMDC | **45.68** | **51.52** | **40.05** | **44.62** | **40.86** | **46.69** | **50.26** | **58.95** |

We've undertaken few-shot learning experiments over four datasets. The term "K-shot" denotes that merely K samples correspond to each category within the training set. Based on the results in the Table 14, topological pre-training consistently enhances the performance of LMs across diverse few-shot scenarios.

### D.6 Experimental Results on Large-scale Text-attributed Graphs

Considering the existence of large text-attributed graphs, we further execute preliminary experiments on the ogbn-papers100M dataset [34] with 111,059,956 nodes and 1,615,685,872 edges. Given the substantial scale of this dataset, which poses challenges for many existing GNNs, we have chosen a set of scalable GNNs (SGC [55], GAMLP [56] and SIGN [57]). Additionally, we have selected several prominent LMs including BERT-Tiny, ELECTRA, and DistilBERT. Different training paradigms are systematically evaluated under the node classification task. Experimental results (accuracy) are demonstrated in the Table 15. GAMLP achieves the best performance among all the GNN models due to its adaptive node-wise feature combination. Furthermore, it is noteworthy that the topological

pre-training (referred to as TPT) methods obtain the SOTA performance over this large dataset, demonstrating the superiority of the proposed pre-training tasks.

Table 15: Node classification results on ogbn-papers100M

| Models | LMs | GNNs | | | | TPT | | | |
|---|---|---|---|---|---|---|---|---|---|
| | PLM | SGC | SIGN | GAMLP | MLP | TMLM | TDK | TCL | TMDC |
| BERT-Tiny | 62.11 | 62.51 | 64.26 | 65.12 | 49.32 | 63.55 | 63.98 | 64.78 | **65.31** |
| ELECTRA-Small | 61.01 | 61.07 | 63.06 | 64.98 | 47.26 | 62.24 | 63.13 | 63.98 | **65.16** |
| DistilBERT | 64.18 | 63.58 | 65.81 | 67.73 | 51.26 | 65.36 | 65.89 | 66.76 | **68.12** |

Table 16: Statics information of the Cresci-2015 and TwiBot-20

| Datasets | User | Tweet | Edge | Human | Bot |
|---|---|---|---|---|---|
| Cresci-2015 | 5,301 | 2,827,757 | 7,086,134 | 1,950 | 3,351 |
| TwiBot-20 | 229,580 | 33,488,192 | 33,716,171 | 5,237 | 6,589 |

## D.7 Text-attributed Graphs in Social Networks

In the realm of text-attributed graphs (TAGs), two prominent and prevalent categories are academic and e-commerce graphs. For example, all text-attributed graphs within the OGB benchmark [34] belong to these two domains. To boost the impact of our benchmark, we have broadened the spectrum of TAG domains by introducing a new dimension, social networks, into our benchmark. To this end, we have incorporated two text-attributed graphs sourced from the widely-used social platform Twitter, named Cresci-2015 [58] and TwiBot-20 [59]. These two datasets are collected for social bot detection. Each node corresponds to a user within Twitter, intrinsically linked to the tweets they have published. The underlying graph topology is shaped by the relationships (e.g., following relations) among users. The labels attributed to each node signify whether the respective user is classified as a bot or not. It is worth noting that, owing to privacy concerns intrinsic to social network data, we provide a summarized representation of the outcomes and insights. The original datasets are available upon request through email, pending the acceptance of this paper. The detailed statistics of the datasets are presented in Table 16.

Table 17: Node classification results on Cresci-2015 and TwiBot-20 dataset (accuracy)

| Datasets | Models | LMs | GNNs | | | Co-Training | | | TPT | | | |
|---|---|---|---|---|---|---|---|---|---|---|---|---|
| | | PLM | GCN | SAGE | GAT | GCN | SAGE | GAT | TMLM | TDK | TCL | TMDC |
| Cresci-2015 | BERT-Tiny | 91.0 | 93.3 | 93.6 | 94.1 | 93.6 | 93.9 | OOM | 91.5 | 92.1 | 93.0 | **94.3** |
| | ELECTRA-Small | 92.0 | 93.0 | 93.4 | 93.9 | OOM | OOM | OOM | 92.3 | 92.9 | 93.8 | **94.6** |
| | DistilBERT | 95.5 | 95.3 | 95.5 | 96.0 | OOM | OOM | OOM | 95.9 | 96.3 | 97.1 | **97.9** |
| | RoBERTa-Base | 97.0 | 96.2 | 96.6 | 96.9 | OOM | OOM | OOM | 97.5 | 97.9 | 98.3 | **98.8** |
| TwiBot-20 | BERT-Tiny | 68.2 | 76.8 | 79.2 | **81.1** | OOM | OOM | OOM | 69.9 | 72.6 | 75.6 | 76.9 |
| | ELECTRA-Small | 69.5 | 75.6 | 78.8 | **80.1** | OOM | OOM | OOM | 71.6 | 73.0 | 76.4 | 77.1 |
| | DistilBERT | 76.5 | 80.9 | 84.6 | **85.4** | OOM | OOM | OOM | 77.6 | 78.1 | 79.3 | 81.2 |
| | RoBERTa-Base | 78.6 | 81.6 | 84.9 | **85.9** | OOM | OOM | OOM | 80.1 | 81.2 | 82.3 | 83.4 |

The node classification results (accuracy) are demonstrated in the Table 17. "OOM" stands for "Out-Of-Memory" on a 32GB V100. On the Cresci-2015 dataset, we observe a noteworthy trend wherein PLM-based methods exhibit comparable or even superior performance compared to GNN-based methods, particularly as the parameters of PLMs increase. RoBERTa consistently surpasses all three GNN models. On the TwiBot-20 dataset, the graph topology seems to be more important. The Graph Attention Network (GAT) emerges as the top-performing model on these datasets. Additionally, the employment of topological pre-training strategies brings performance enhancements for both datasets, underscoring their efficacy in advancing model capabilities.

## D.8 Evaluations of LLMs on TAGs

In our benchmark, we have mainly studied PLMs based on encoder architecture like BERT [10]. However, most of the recent rapidly developing models in the NLP field are LLMs based on the

Decoder architecture represented by GPT [32]. Following the experiment workflow denoted in TAPE [6], we leverage LLMs to generate high-quality node features for TAGs. From the perspective of LMs, we incorporate the recent and prominent large language models as the baselines, including T5 (11B) [60], LaMDA (137B) [61], GPT-3 (175B) [28] and PaLM (540B) [62]. In brief, we use LLMs' inference APIs to generate explanations about the original text. These explanations are incorporated into the original text for fine-tuning the respective LMs. Finally, we extract features from LMs and use them to train the downstream GNNs. DistilBERT is selected as the feature extractor LMs, and GCN, GAT, and SAGE are selected as the downstream GNNs. Experimental results are presented in the Table 18. PaLM has consistently attained the most impressive performance across all downstream GNNs and datasets.

Table 18: Node classification experiments on the three datasets. The row "LLMs" denotes using LLMs to generate explanations about the raw text to fine-tune the LM and generate node features for downstream GNNs. The row "Raw" denotes using the original LMs to generate node features.

|  | Methods | Arxiv | | | Children | | | Photo | | |
|---|---|---|---|---|---|---|---|---|---|---|
|  |  | GCN | GAT | SAGE | GCN | GAT | SAGE | GCN | GAT | SAGE |
| Raw | DistilBERT | 73.39 | 73.48 | 74.48 | 58.19 | 57.91 | 59.33 | 82.91 | 83.75 | 83.50 |
| LLMs | T5 | 73.51 | 73.80 | 74.22 | 58.61 | 58.14 | 59.12 | 83.76 | 84.75 | 84.33 |
|  | LaMDA | 74.06 | 74.55 | 74.92 | 59.67 | 59.56 | 60.86 | 84.23 | 84.95 | 84.59 |
|  | GPT-3 | 74.41 | 74.81 | 75.34 | 60.12 | 60.01 | 61.12 | 84.56 | 85.23 | 84.78 |
|  | PaLM | **75.22** | **76.43** | **76.72** | **61.59** | **61.26** | **62.23** | **85.45** | **85.95** | **85.69** |

## D.9 Study on the Selection of Node Attributes

In our previous experiments, we have observed a phenomenon that LMs usually perform much better on datasets that use product descriptions as text attributes than those that use product reviews. In order to be more explicit about the effect of node attribute selection on different models, we have reconstructed two datasets with product description as text attribute for Ele-Photo and Ele-Computers, where LMs performed poorly before. In the Table 19, datasets labeled with "RW" incorporate user reviews as node attributes, while those marked "DS" employ product descriptions as attributes. Different training paradigms are systematically evaluated under the node classification task. BERT-Base is selected as the foundational text encoder model. Evidently, the performance of PLMs when utilizing descriptions as node attributes demonstrates a substantial enhancement in contrast to reviews. This observation underscores the pivotal role that the selection of node attributes plays in achieving desirable TAG representation learning.

Table 19: Experimental results (accuracy) of node classification on datasets with different text attributes.

|  | LMs | GNNs | | | TPT | | | |
|---|---|---|---|---|---|---|---|---|
| Datasets | PLM | SAGE | GCN | GAT | TMLM | TDK | TCL | TMDC |
| Photo-RW | 77.53 | 83.27 | 82.70 | 83.74 | 78.54 | 81.04 | 82.85 | 84.09 |
| Photo-DS | 85.07 | 84.86 | 83.72 | 85.16 | 86.15 | 86.49 | 87.26 | 88.15 |
| Computers-RW | 61.96 | 88.30 | 87.86 | 88.63 | 63.43 | 82.85 | 64.93 | 69.45 |
| Computers-DS | 86.41 | 88.90 | 88.26 | 89.13 | 87.56 | 87.26 | 88.96 | 89.53 |

# E   Broader Impact

Representation learning on text-attributed graphs is a fast-growing and promising research field, and covers a wide range of applications. We start this benchmark to call more researchers' attention to this common data type. The proposed benchmark CS-TAG can significantly facilitate the development of the textual-attributed graph learning. CS-TAG deeply and extensively explores the paradigm of combining pre-trained language models (PLMs) with graph neural networks (GNNs), and provides a comprehensive evaluation over multiple large constructed datasets. Nevertheless, there are still

lots of research gaps need to be bridged. First, the self-supervised and unsupervised learning in textual-attributed graphs are not included in CS-TAG, which play an important role in graph data mining research. Second, CS-TAG does not pay much attention to link prediction task that has many applications in real world, such as recommendation system and drug discovery. Third, the interpretability of textual-attributed graph learning is not discussed here. Comparing with the vectorized features in other graphs, such as chemical molecules, the text attributes can be directly understood by human. Therefore, the textual-attributed graphs are more human-intelligible and have promising potential for interpretability research.

In the future, we will keep track on the newly emerged problems in textual-attributed graphs and provide more solid experimental results and detailed analyses to improve CS-TAG consistently. It is an ongoing effort and we strive to continuously include more datasets and evaluate different methods to advance the field.

