# OpenReview forum: "A Comprehensive Study on Text-attributed Graphs: Benchmarking and Rethinking"
_NeurIPS.cc/2023/Track/Datasets_and_Benchmarks — NeurIPS 2023 Datasets and Benchmarks Poster_

### Official Review · Reviewer_fbKv · 2023-07-20

**Rating:** 6
**Confidence:** 3
**Correctness:** Yes
**Clarity:** Yes

**Strengths:**

1. The proposed benchmarks are more large-scale than traditional datasets, with millions of nodes and edges.
2. The authors have tested various LLMs to better model the semantic information.

**Additional Feedback:**

See "Opportunities For Improvement“

**Documentation:**

Could be improved

**Limitations:**

Yes

**Opportunities For Improvement:**

1. There are existing large-scale text-attributed graphs, such as ogbn-papers100M, which are even larger than the proposed benchmark. It would be helpful to know if the authors have considered incorporating these datasets.
2. While the authors provide 8 datasets, they are only from 2 domains (academic and e-commerce), which may limit the impact and generalization ability of the benchmark.
3. The dataset page could benefit from improved documentation. For example, it is not immediately clear how users can test their own GNN models.

**Relation To Prior Work:**

Could be improved

**Summary And Contributions:**

The paper proposes a benchmark for text-attributed graphs, using 8 large-scale datasets and testing various GNN models. The benchmarks are larger in scale than traditional datasets and the authors have tested various LLMs. The authors also make a few observations from the empirical results.

---

> ### Author Response · Authors · 2023-08-24
> **Response to reviewer fbKv (Part 1)**
>
> >Q1: There are existing large-scale text-attributed graphs, such as ogbn-papers100M, which are even larger than the proposed benchmark. It would be helpful to know if the authors have considered incorporating these datasets.
>
> Thanks for your valuable suggestion.  We have successfully incorporated the ogbn-papers100M dataset into our benchmark and executed preliminary experiments accordingly. Given the substantial scale of this dataset, which poses challenges for many existing GNNs, we have chosen a set of scalable GNNs (SGC[1], GAMLP[2], and SIGN[3]). Additionally, we have selected several prominent LMs including BERT-Tiny, ELECTRA, and DistilBERT. Different training paradigms are systematically evaluated under the node classification task. Experimental results (accuracy) are demonstrated in the following table. GAMLP achieves the best performance among all the GNN models due to its adaptive node-wise feature combination [2]. Furthermore, it is noteworthy that the topological pre-training (referred to as TPT) methods obtain the SOTA performance over this large dataset, demonstrating the superiority of the proposed pre-training tasks. We have detailed these results in the revised Appendix D.6.
>
> | Papers100M     | | BERT-Tiny | ELECTRA | DistilBERT |
> | ---- | ---------- | -------- | ------- | ---------- |
> | **LMs**  | PLM        | 62.11    | 61.01   | 64.18      |
> | **GNNs** | SGC        | 62.51    | 61.07   | 63.58      |
> |      | SIGN       | 64.26    | 63.06   | 65.81      |
> |      | GAMLP      | 65.12    | 64.98   | 67.73      |
> |      | MLP        | 49.32    | 47.26   | 51.26      |
> | **TPT**  | TMLM       | 63.55    | 62.24   | 65.36      |
> |      | TDK        | 63.98    | 63.13   | 65.89      |
> |      | TCL        | 64.78    | 63.98   | 66.76      |
> |      | TMDC       | **65.31**    | **65.16**   | **68.12**      |
>
> [1] Simplifying Graph Convolutional Networks. ICML 2019
>
> [2] Graph Attention Multi-Layer Perceptron. KDD 2022
>
> [3] SIGN: Scalable Inception Graph Neural Networks.  CoRR abs/2004.11198

---

> > ### Author Response · Authors · 2023-08-24
> > **Response to reviewer fbKv (Part 2)**
> >
> > > Q2: While the authors provide 8 datasets, they are only from 2 domains (academic and e-commerce), which may limit the impact and generalization ability of the benchmark.
> >
> > We deeply appreciate your insightful comment. In the realm of text-attributed graphs (TAGs), two prominent and prevalent categories are academic and e-commerce graphs.  For example, all text-attributed graphs within the OGB benchmark belong to these two domains. Following your invaluable suggestion, we have broadened the spectrum of TAG domains by introducing a new dimension, social networks, into our benchmark. To this end, we have incorporated two text-attributed graphs sourced from the widely-used social platform Twitter, named Cresci-2015 [1] and TwiBot-20 [2]. These two datasets are collected for social bot detection. Each node corresponds to a user within Twitter, intrinsically linked to the tweets they have published. The underlying graph topology is shaped by the relationships (e.g., following relations) among users. The labels attributed to each node signify whether the respective user is classified as a bot or not. It is worth noting that, owing to privacy concerns intrinsic to social network data, we provide a summarized representation of the outcomes and insights. The original datasets are available upon request through email, pending the acceptance of this paper. The detailed statistics of the datasets are as follows. These two social networks have been added and discussed in the revised Appendix D.7.
> >
> > | Datasets | Cresci-2015     | TwiBot-20 |
> > | -------- | ------- | --------- |
> > | #User     | 5,301    | 229,580    |
> > | #Tweet    | 2,827,757 | 33,488,192  |
> > | #Edge     | 7,086,134 | 33,716,171  |
> > | #Human    | 1,950    | 5,237      |
> > | #Bot      | 3,351    | 6,589      |
> >
> > The node classification results (accuracy) are demonstrated in the following table. 'OOM' stands for 'Out-Of-Memory' on a 32GB V100.  On the Cresci-2015 dataset, we observe a noteworthy trend wherein PLM-based methods exhibit comparable or even superior performance compared to GNN-based methods, particularly as the parameters of PLMs increase. RoBERTa consistently surpasses all three GNN models. On the TwiBot-20 dataset, the graph topology seems to be more important. The Graph Attention Network (GAT) emerges as the top-performing model on these datasets. Additionally, the employment of topological pre-training strategies brings performance enhancements for both datasets, underscoring their efficacy in advancing model capabilities.
> >
> >
> > | Cresci-2015           |   | BERT-Tiny | ELECTRA | DistilBERT | RoBERTa |
> > | ----------- | ---- | -------- | ------- | ---------- | ------- |
> > | **LMs**         | PLM  | 91.0     | 92.0    | 95.5       | 97.0    |
> > | **GNNs**        | GCN  | 93.3     | 93.0    | 95.3       | 96.2    |
> > |             | SAGE | 93.6     | 93.4    | 95.5       | 96.6    |
> > |             | GAT  | 94.1     | 93.9    | 96.0       | 96.9    |
> > | **Co-Training** | GCN  | 93.6     | OOM     | OOM        | OOM     |
> > |             | SAGE | 93.9     | OOM     | OOM        | OOM     |
> > |             | GAT  | OOM      | OOM     | OOM        | OOM     |
> > | **TPT**        | TMLM | 91.5     | 92.3    | 95.9       | 97.5    |
> > |             | TDK  | 92.1     | 92.9    | 96.3       | 97.9    |
> > |             | TCL  | 93.0     | 93.8    | 97.1       | 98.3    |
> > |             | TMDC | **94.3**    | **94.6**    | **97.9**       | **98.8**    |
> >
> >
> > | TwiBot-20            |  | BERT-Tiny | ELECTRA | DistilBERT | RoBERTa |
> > | ----------- | --------- | -------- | ------- | ---------- | ------- |
> > | **LMs**         | PLM       | 68.2     | 69.5    | 76.5       | 78.6    |
> > | **GNNs**        | GCN       | 76.8     | 75.6    | 80.9       | 81.6    |
> > |             | SAGE      | 79.2     | 78.8    | 84.6       | 84.9    |
> > |             | GAT       | **81.1**     | **80.1**    | **85.4**       | **85.9**    |
> > | **TPT**         | TMLM      | 69.9     | 71.6    | 77.6       | 80.1    |
> > |             | TDK       | 72.6     | 73.0    | 78.1       | 81.2    |
> > |             | TCL       | 75.6     | 76.4    | 79.3       | 82.3    |
> > |             | TMDC      | 76.9     | 77.1    | 81.2       | 83.4    |
> >
> >
> > [1] Fame for sale: Effificient detection of fake twitter followers.  Decision Support Systems 2015
> >
> > [2] TwiBot-20: A Comprehensive Twitter Bot Detection Benchmark. CIKM 2021

---

> ### Author Response · Authors · 2023-08-24
> **Response to reviewer fbKv (Part 3)**
>
> >Q3: The dataset page could benefit from improved documentation. For example, it is not immediately clear how users can test their own GNN models.
>
> We greatly appreciate your insightful suggestions. Following your guidance, we have made the necessary revisions to the code documentation within our GitHub repository. Specifically, users are now able to execute the following steps in order to conduct testing of their GNN models:
>
> 1. Navigate to the file GNN/model/GNN_library.py and proceed to define a personalized GNN model, drawing inspiration from the existing models such as GCN and GAT that are present in the codebase.
>
> 2. Access GNN/GNN_args.py and locate the args_init() function, where one can include model-specific parameters tailored to their newly crafted model.
>
> 3. Within GNN/GNN.py, import the newly designed model and accordingly update the gen_model() function to reflect the incorporation of the model.
>
>
> To facilitate a seamless testing experience for users, we have streamlined the above process within the test_model.sh script. This means that users can conveniently evaluate their models through the command 'bash test_model.sh model_name data_name'. For a comprehensive walkthrough of this process, a detailed set of instructions is available at the following URL: [https://github.com/sktsherlock/TAG-Benchmark](https://github.com/sktsherlock/TAG-Benchmark).

---

> > ### Comment · Reviewer_fbKv · 2023-08-25
> >
> > Thanks for the response and updating of the dataset and codebase. I have increased my score accordingly.

---

> > > ### Author Response · Authors · 2023-08-25
> > > **Thank you for your acknowledgment and any further discussions are welcomed**
> > >
> > > Dear reviewer fbKv,
> > >
> > > Thank you very much for your reply!  Your valuable comments and suggestions help improve our paper a lot! Further discussions are welcomed at any time. We would be happy to have insightful discussions with you and make further improvements.
> > >
> > > Sincerely,
> > >
> > > Authors

---

### Official Review · Reviewer_ZLyP · 2023-07-21
**A Comprehensive Study on Text-attributed Graphs: Benchmarking and Rethinking**

**Rating:** 6
**Confidence:** 4
**Correctness:** Yes, the dataset is constructed in a …
**Clarity:** Yes, the paper is overall well writte…

**Strengths:**

1. The proposed dataset are comprehensive, scalable and with raw text.
2. The problem co-training GNN and LMs are interesting and useful.
3. The authors have some interesting findings and may help the research field.

**Additional Feedback:**

Can authors give more detailed explaination on the findings, especially on the co-training results?

**Documentation:**

Yes, the details are provided.

**Ethics:**

Not appliable.

**Limitations:**

The findings may lack some sufficient explaination. For example, the co-training framework performs worse and the authors think "This is because jointly optimizing GNNs and PLMs can be more difficult than training the respective models separately." If so, are the joint optimization well conducted and farily compared?

**Opportunities For Improvement:**

1. The type of proposed dataset is somehow narrow, for example, the social media such as twitter and Weibo.
2. The GNNs and LMs used in this paper is somehow trival.


**Relation To Prior Work:**

The major difference lies on the raw text associated with nodes, which is critical for LM training.

**Summary And Contributions:**

This paper studies the problem of representation learning on Text-attributed graphs (TAGs).
This paper propose a set of large scale text-attributed graph dataset with raw text.
The authors conduct experiments on the datasets and have some interesting findings.

---

> ### Author Response · Authors · 2023-08-24
> **Response to reviewer ZLyP (Part 1)**
>
> > Q1: The type of proposed dataset is somehow narrow, for example, the social media such as twitter and Weibo.
>
> We deeply appreciate your insightful comment. In the realm of text-attributed graphs (TAGs), two prominent and prevalent categories are academic and e-commerce graphs.  For example, all text-attributed graphs within the OGB benchmark belong to these two domains. Following your invaluable suggestion, we have broadened the spectrum of TAG domains by introducing a new dimension, social networks, into our benchmark. To this end, we have incorporated two text-attributed graphs sourced from the widely-used social platform Twitter, named Cresci-2015 [1] and TwiBot-20 [2]. These two datasets are collected for social bot detection. Each node corresponds to a user within Twitter, intrinsically linked to the tweets they have published. The underlying graph topology is shaped by the relationships (e.g., following relations) among users. The labels attributed to each node signify whether the respective user is classified as a bot or not. It is worth noting that, owing to privacy concerns intrinsic to social network data, we provide a summarized representation of the outcomes and insights. The original datasets are available upon request through email, pending the acceptance of this paper. The detailed statistics of the datasets are as follows. These two social networks have been added and discussed in the revised Appendix D.7.
>
> | Datasets | Cresci-2015     | TwiBot-20 |
> | -------- | ------- | --------- |
> | #User     | 5,301    | 229,580    |
> | #Tweet    | 2,827,757 | 33,488,192  |
> | #Edge     | 7,086,134 | 33,716,171  |
> | #Human    | 1,950    | 5,237      |
> | #Bot      | 3,351    | 6,589      |
>
> The node classification results (accuracy) are demonstrated in the following table. 'OOM' stands for 'Out-Of-Memory' on a 32GB V100.  On the Cresci-2015 dataset, we observe a noteworthy trend wherein PLM-based methods exhibit comparable or even superior performance compared to GNN-based methods, particularly as the parameters of PLMs increase. RoBERTa consistently surpasses all three GNN models. On the TwiBot-20 dataset, the graph topology seems to be more important. The Graph Attention Network (GAT) emerges as the top-performing model on these datasets. Additionally, the employment of topological pre-training strategies brings performance enhancements for both datasets, underscoring their efficacy in advancing model capabilities.
>
>
> | Cresci-2015           |   | BERT-Tiny | ELECTRA | DistilBERT | RoBERTa |
> | ----------- | ---- | -------- | ------- | ---------- | ------- |
> | **LMs**         | PLM  | 91.0     | 92.0    | 95.5       | 97.0    |
> | **GNNs**        | GCN  | 93.3     | 93.0    | 95.3       | 96.2    |
> |             | SAGE | 93.6     | 93.4    | 95.5       | 96.6    |
> |             | GAT  | 94.1     | 93.9    | 96.0       | 96.9    |
> | **Co-Training** | GCN  | 93.6     | OOM     | OOM        | OOM     |
> |             | SAGE | 93.9     | OOM     | OOM        | OOM     |
> |             | GAT  | OOM      | OOM     | OOM        | OOM     |
> | **TPT**        | TMLM | 91.5     | 92.3    | 95.9       | 97.5    |
> |             | TDK  | 92.1     | 92.9    | 96.3       | 97.9    |
> |             | TCL  | 93.0     | 93.8    | 97.1       | 98.3    |
> |             | TMDC | **94.3**    | **94.6**    | **97.9**       | **98.8**    |
>
>
> | TwiBot-20            |  | BERT-Tiny | ELECTRA | DistilBERT | RoBERTa |
> | ----------- | --------- | -------- | ------- | ---------- | ------- |
> | **LMs**         | PLM       | 68.2     | 69.5    | 76.5       | 78.6    |
> | **GNNs**        | GCN       | 76.8     | 75.6    | 80.9       | 81.6    |
> |             | SAGE      | 79.2     | 78.8    | 84.6       | 84.9    |
> |             | GAT       | **81.1**     | **80.1**    | **85.4**       | **85.9**    |
> | **TPT**         | TMLM      | 69.9     | 71.6    | 77.6       | 80.1    |
> |             | TDK       | 72.6     | 73.0    | 78.1       | 81.2    |
> |             | TCL       | 75.6     | 76.4    | 79.3       | 82.3    |
> |             | TMDC      | 76.9     | 77.1    | 81.2       | 83.4    |
>
>
> [1] Fame for sale: Effificient detection of fake twitter followers.  Decision Support Systems 2015
>
> [2] TwiBot-20: A Comprehensive Twitter Bot Detection Benchmark. CIKM 2021

---

> ### Author Response · Authors · 2023-08-24
> **Response to reviewer ZLyP (Part 2)**
>
> >Q2: The GNNs and LMs used in this paper is somehow trival.
>
> We appreciate your valuable suggestion. Within the manuscript, the selection of seven GNNs and five LMs with different scales is grounded in their widespread popularity and significant influence within their respective domains. These models are commonly employed in related benchmarks such as ogbn-arxiv. Our primary objective is to present a comprehensive compilation of results and analyses pertaining to these widely recognized methodologies. By doing so, we strive to enhance the practicality and viability of this benchmark. Building upon your insightful comment, we have incorporated additional recent GNNs and LMs into the study for the purpose of comparison.
>
> 1) Two recent GNN models, NodeFormer[1] and RevGAT[2] are added as the new GNN methods.  NodeFormer is a scalable graph Transformer for large-scale graphs, which achieves all-pair message passing with linear complexity. RevGAT combines reversible connectivity with a deep network architecture to form a deep and efficient GNN. Experimental results are presented in the following table. DistilBERT serves the purpose of encoding textual content contained within the nodes. One can observe that RevGAT achieves the best performance across all datasets. This is mainly due to its deep reversible architecture, which allows the model to capture deeper and wider topological information. More detailed experimental results can be found in Appendix D.1 of the revised version.
>
> | Datasets  | GCN   | GAT   | SAGE  | RevGAT | NodeFormer |
> | --------- | ----- | ----- | ----- | ------ | ---------- |
> | Arxiv     | 73.39 | 73.48 | 74.48 | **74.68**  | 73.56      |
> | Children  | 58.19 | 57.91 | 59.33 | **59.28**  | 58.03      |
> | History   | 85.14 | 84.68 | 85.56 | **85.67**  | 84.49      |
> | Photo     | 82.91 | 83.75 | 83.50  | **84.34**  | 82.69      |
> | Computers | 88.37 | 89.07 | 88.67 | **89.32**  | 87.96      |
> | Sports    | 86.15 | 86.58 | 87.51 | **87.62**  | 85.95      |
>
> 2) From the perspective of LMs, we incorporate the recent and prominent large language models as the baselines, including T5 (11B)[3], LaMDA (137B)[4], GPT-3 (175B)[5] and PaLM (540B)[6]. Following the experiment workflow denoted in TAPE [7], we leverage LLMs to generate high-quality node features for TAGs.  In brief, we use LLMs' inference APIs to generate explanations about the original text. These explanations are incorporated into the original text for fine-tuning the respective LMs. Finally, we extract features from LMs and use them to train the downstream GNNs. DistilBERT is selected as the feature extractor LMs, and GCN, GAT, and SAGE are selected as the downstream GNNs. Experimental results are presented in the following tables. PaLM has consistently attained the most impressive performance across all downstream GNNs and datasets. We have added the details of these updates in the revised Appendix D.8.
>
>
> | Arxiv |       | GCN  | GAT   | SAGE  |
> | ----- | ----- | ---- | ----- | ----- |
> | Raw   | DistilBERT  | 73.39 | 73.48  | 74.48 |
> | LLMs  | T5    | 73.51 | 73.80  | 74.22 |
> |       | LaMDA | 74.06 | 74.55 | 74.92 |
> |       | GPT-3 | 74.41 | 74.81 | 75.34 |
> |       | PaLM  | 75.22 | 76.43 | 76.72 |
>
>
> | Children |       | GCN   | GAT   | SAGE  |
> | -------- | ----- | ----- | ----- | ----- |
> | Raw      | DistilBERT  | 58.19 | 57.91 | 59.33 |
> | LLMs     | T5    | 58.61 | 58.14 | 59.12 |
> |          | LaMDA | 59.67 | 59.56 | 60.86 |
> |          | GPT-3 | 60.12 | 60.01 | 61.12 |
> |          | PaLM  | 61.59 | 61.26 | 62.23 |
>
>
> | Photo |       | GCN   | GAT   | SAGE  |
> | ----- | ----- | ----- | ----- | ----- |
> | Raw   | DistilBERT  | 82.91 | 83.75 | 83.50 |
> | LLMs  | T5    | 83.76 | 84.75 | 84.33 |
> |       | LaMDA | 84.23 | 84.95 | 84.59 |
> |       | GPT-3 | 84.56 | 85.23 | 84.78 |
> |       | PaLM  | 85.45 | 85.95 | 85.69 |
>
>
> [1] Training Graph Neural Networks with 1000 Layers.  ICML 2021
>
> [2] NodeFormer: A Scalable Graph Structure Learning Transformer for Node Classification. NeurIPS 2022
>
> [3] Exploring the Limits of Transfer Learning with a Unified Text-to-Text Transformer.  J. Mach. Learn. Res. 21
>
> [4] LaMDA: Language Models for Dialog Applications. CoRR abs/2201.08239
>
> [5] Language models are Few-Shot Learners. NeurIPS 2020
>
> [6] PaLM: Scaling Language Modeling with Pathways. CoRR abs/2204.02311
>
> [7] Explanations as Features: LLM-Based Features for Text-Attributed Graphs. CoRR abs/2305.19523

---

> ### Author Response · Authors · 2023-08-24
> **Response to reviewer ZLyP (Part 3)**
>
> > Q3: The findings may lack some sufficient explaination. For example, the co-training framework performs worse and the authors think "This is because jointly optimizing GNNs and PLMs can be more difficult than training the respective models separately.". Can authors give more detailed explaination on the findings, especially on the co-training results?
>
> Thanks for pointing out this question. The co-training framework requires simultaneous training of PLMs and GNNs. For example,  given k neighbors for each node, co-training methods need to simultaneously learn (k + 1) embeddings through pre-trained language models (e.g., BERT) and aggregate these embeddings by the GNNs, leading to additional computation complexity and substantial memory costs. Therefore, the memory requirement and time cost of co-training paradigms are significantly increased.
>
> In order to facilitate the co-training of PLMs and GNNs, a commonly employed approach is to reduce either the batch size (bsz) or the number of neighbors (k) utilized per layer. However, it's important to note that decreasing the number of neighbors results in a loss of crucial topological information, ultimately leading to degraded performance [1]. Meanwhile, decreasing the batch size leads to prolonged training times [2]. We dedicated significant effort using our resources (32GB V100) to conduct numerous trials involving hyper-parameter grid searches. However, we encountered instances of training instability and suboptimal performance of the co-training paradigm across certain models and datasets. As a result, while co-training remains theoretically advantageous, considering both efficiency and effectiveness, we can only conclude that this strategy holds a position of comparability rather than being definitively optimal.
>
> [1] Training Graph Neural Networks with 1000 Layers.  ICML 2021
>
> [2] On Efficient Training of Large-Scale Deep Learning Models: A Literature Review. CoRR abs/2304.03589.

---

> > ### Author Response · Authors · 2023-08-28
> >
> > Dear reviewer ZLyP,
> >
> > Thanks again for your insightful comments, which, we believe, are very important to improve our paper. In the rebuttal, we have tried to answer your questions one by one. If you have further questions, we are very happy to discuss them. Thanks.
> >
> > Sincerely,
> >
> > Authors

---

### Official Review · Reviewer_ghFj · 2023-07-21
**good paper, accept (The work is there but needs to be better presented)**

**Rating:** 8
**Confidence:** 5

**Strengths:**

**[Significance of the contribution]**
High. See above in [Contributions].

**[Relevance to the broader research community]**
High.

**[Quality of the research]**
In general the quality of this research is high.

**[Ethical and social implications]**
NA at this stage.

**Additional Feedback:**

1. Double check the formatting, e.g., l. 310, Section 4.4 should not end with a dot (.), apart from the comments in the "Clarity" section.
2. I briefly went over the appendix and found many details. Some corrections:
    1. Section A.2, $h$ is used in the main text, and $z$ in appendix, please make them consistent.
    2. I have suggested merging the tables into a bigger one, also consider doing so for Tables 3-16, based on various metrics, of course.
    3. Any reason why a table like Table 3 is not produced for the link prediction task?

**Clarity:**

**[Structure]**
Clear.

**[Text]**
1. General remark about footnotes. They should always come after punctuation and without spacing in between, e.g, line 188.

**[Table]**
1. Table 1: Add the tasks of node classification and link predictions to the datasets.
2. Write the tables in the format of Table X: line 277, (Table 2), line 301 (Table 4)
3. Report results like Table 2 for all the datasets in a bigger table. For instance, you can list some important findings with numbers in the main text and leave the big table to appendix. The current structure makes it hard to find all the results in one place. I find it hard to jump between tables and compare the performances.
4. Table 2: Any reasons why other PLM based methods like Roberta are not tested? Make sure you indicate what "scale" means here (in appendix for instance), as it might not be clear to all readers that "scale" means the different versions of PLMs (number of parameters).
Please clarify what "diff" means. It is the $\delta$ between the best and worse performers for one model (column). I would suggest to also mention the bold numbers are for each model (row)

5. Table 3: Maybe it would be clearer to mark the highest number per row (dataset) bold and mark boundaries between PLM based, GNN based and Co-training based.
Also for me, it is counter intuitive to mark the increase with red and decrease with green.

6. Table 4: It makes sense to add a vertical line between the datasets. There are only four datasets shown here, what about the other two on node classification?
    - If I am not mistaken, 84.15 (Large-Roberta-Photo-GNNs) is the best performer on the dataset Photo, which is still worse compared to 86.64 (Photo-Co-Training based) in Table 3.
    - I would suggest the authors to rethink the table strutures of Tables 2, 3, 4. It would make more sense to put them into one big table as the results are interlinked and it is easier to compare.
    - If you want to keep the structure like Table 4, please at least mention that the GNN column is the best performers of GNN methods, and you can do the same for PLMs (add a PLM column) and co-training (add a co-training column and report the best GNN variant for each PLM variant).
    - The number in subscript is computed wrt the lowest number in row, please mention that and rethink the coloring.
    - Also it would be interesting to comment on the columns, in general, the larger the scale of an LLM is, the better the performance is. But it is not always the case, see column (History-GNNs), the best performer is Small-DistilBert, 85.56.

**[Figure]**
1. Figure 2: There are eight datasets, of which we only see 6 points. Please clarify.
2. Figure 4: It is not clear what "Fanout" means. But I guess it is supposed to be the number of neighbors in the x-axis.
Please add markers to the lines, as the colors are not black-white-print friendly.

**Correctness:**

**[Claims]**
 In general ok.

**[Dataset constructed in a sound way?] **
In general ok.
In Appendix B.3 the authors introduce the construction of datasets. I have a few comments which could either fall under the "Correctness" or "Clarity".
1. l.114: "three-level label" for each book, does the node classification classify the hierarchy as well, or they are flattened, or just the lowest level of labels? Same question for l.122 on dataset Ele-Computer/Photo.
2. The authors mention in Section 4.3 that the performance of TMDC is bad on the dataset Photo because the text on each node is the review. Can we use the product description as node text attributes, or that is not an option?
3. Same question for Sports-Fitness, what is the text attribute?
4. Why two different metrics for CitationV8 and GoodReads? MRR and Hits@K, why not evaluate both for these two datasets?
5. For CitationV8 and GoodReads, you mention negative examples (2k and 5k, repsectively). Are these the number (e.g., 2k) wrt to each item (or book) or in total there are 2k negative examples that do not get referenced by any of the nodes in the graph? Is it a process during data set construction or is it a process when evaluating the models (aka computing MRR)?

**[Benchmark evaluation methods and experiment design appropriate and performed correctly?]**
See the comments above.

**Documentation:**

**[Datasets with sufficient detail on data collection and organization, availability and maintenance, and ethical and responsible use? ]**
- In general good quality.
- Code documentation. This is not clear from the codebase how to run the benchmark, the authors need to specify scripts to call and also add a dir struture, which gives an overview of the scripts.

**[Dataset submissions include documentation and intended uses; a URL for reviewer access to the dataset; and a hosting, licensing, and maintenance plan? ]**
Yes.

**[Benchmarks with sufficient detail to support reproducibility?]**
Yes.

**Ethics:**

**[Concerns are discussed?]** NA.

**[Datasets' consideration of consent and privacy, responsible use, and legal compliance?]** NA.


**Limitations:**

**[Addressed the limitations and potential negative societal impact of their work]**
NA at this stage.

**[Constructive suggestions for improvement]**
1. Many details are in appendix, which is absolutely fine, as long as the authors indicate clearly where to find additional details via coreference.
Many of these coreferences are missing.
    1. Mention link prediction results are in Appendix X, for instance in the paragraph before the conclusion.
    2. Mention results on other datasets (like Table 2) are in Appendix X, l. 276-277.
    3. Mention how TMDC is done briefly and link the details to the Appendix X, l. 320.

2. Code documentation. This is not clear from the codebase how to run the benchmark, the authors need to specify scripts to call and also add a dir structure, which gives an overview of the scripts.

**[Potential points to include]**
I would like to see the discussions on efficiency and scalability (as mentioned above).

**Opportunities For Improvement:**

**[Significance of the contribution]**
1. The authors can mention how this line of research can be used, e.g., understanding recommender systems (products, citations).
This can help the industries to situate themselves and also actively participate the benchmarking.

2. I would like to see the discussions on efficiency and scalability. Since the authors mentioned that co-training of GNN and PLM is costly, how much improvement of time and resources are we talking about if we switch to topological PLM?
These numbers are missing from the current manuscript.

**[Relevance to the broader research community]**
1. The authors can mention how this line of research can be used, e.g., understanding recommender systems (products, citations).

**[Quality of the research]**
The authors could work on the results presentations as I will elaborate later.

**Relation To Prior Work:**

**[Review of prior work]**
Good.

**[Is it clearly discussed how this work differs from previous contributions?]**
 Yes.


**Summary And Contributions:**

**[Summary]**
The paper presents a comprehensive study on text-attributed graphs (TAG) with eight new datasets and a thorough benchmarking between PLMs, GNNs, their co-training, and topological pre-trained LMs. The authors have reviewed the necessity of having such a TAG benchmark from the perspective of data sets, model efficiency and scalability.

**[Contributions]**
- (C1): A new benchmark on TAG (data and models).
- (C2): Extensive experiments and results which hint on very interesting future directions.
- (C3): Personally I also work on the intersection between GNN and PLM. It would be very exciting to see the synergy of these two fields of research.

---

> ### Author Response · Authors · 2023-08-24
> **Response to reviewer ghFj (Part 1)**
>
> **Opportunities For Improvement**:
>
> >Q1: The authors can mention how this line of research can be used, e.g., understanding recommender systems (products, citations). This can help the industries to situate themselves and also actively participate the benchmarking.
>
> A1: Thank you for your valuable suggestion. Text-attributed graphs have emerged as a prominent graph format, which finds extensive applications in modeling real-world tasks, such as the mentioned recommender systems. Our research concentrates on achieving a comprehensive understanding of the textual attributes embedded within a single node and the topological structural connections between nodes. For example, a famous example in recommender systems is the association between "diaper" and "beer", commonly co-purchased by customers, thereby establishing links between these items in the item-item graph. To achieve the optimal item representation, a prerequisite is to capture the inherent characteristics of a given item by modeling its metadata, such as title and descriptions. Simultaneously, it is imperative to incorporate valuable and unique signals derived from the graph's topological connections into the representation learning process. Given that real-world graph topology is usually shaped by human behaviors, there exists unique human perceptions and knowledge in the topology beyond the pure semantics (e.g., 'diaper' and 'beer' are semantically different but are connected in the co-purchased graph). Consequently, it is imperative to delve into the effective and efficient fusion of intrinsic semantics within individual nodes and the topological connections among different nodes on the text-attributed graphs. Moreover, the scope of our research extends beyond recommender systems [1], encompassing domains like leveraging user behaviors to enhance the performance of on-line sponsored search, including AdsGNN [2], HBGLR [3], and PASS [4]. We have added more discussions and explanations on the impact of our work on different industry domains in the revision.
>
> [1] Generative Recommendation: Towards Next-generation Recommender Paradigm. CoRR 2304.03516
>
> [2] AdsGNN: Behavior-Graph Augmented Relevance Modeling in Sponsored Search.  SIGIR 2021
>
> [3] Improving Relevance Modeling via Heterogeneous Behavior Graph Learning in Bing Ads. KDD 2022
>
> [4] PASS: Personalized Advertiser-aware Sponsored Search. KDD 2023
>
> >Q2: I would like to see the discussions on efficiency and scalability. Since the authors mentioned that co-training of GNN and PLM is costly, how much improvement of time and resources are we talking about if we switch to topological PLM?
>
> A2: Thanks for your insightful comments. We present the accuracy, GPU memory cost on a single 32GB V100, and the total training time of the co-training paradigm and topological PLMs across four datasets. BERT-Tiny+SAGE and BERT-Base+SAGE are selected as the co-training approaches. As one of the best topological pre-training tasks, topological contrastive learning (TCL) is selected to enhance the PLMs. Experimental results are presented in the following table. For the small language model BERT-Tiny, the co-training paradigm costs more memories (\~4X) compared to the TCL, and is much slower (\~6X) than TCL. If we use the larger language model like BERT-Base, co-training models will be out-of-memory in a single 32GB V100. Thus, the proposed topological pre-training paradigm is more efficient and practical than the co-training ones. We have added the detailed discussions on efficiency and scalability in the revised Appendix D.3.
>
> | BERT-Tiny   | | Co-Training        |            |    |   TCL     |            |
> | ---------- | ----------- | ------ | ---------- | ----- | ------ | ---------- |
> |            | **Acc**         | **Memory** | **Time (min)** | **Acc**   | **Memory** | **Time (min)** |
> | Arxiv      | 73.57       | 76.27% | 44.0       | 71.80  | 27.59% | 7.0        |
> | Children   | 59.70        | 97.28% | 15.5       | 54.11 | 19.76% | 2.0        |
> | History    | 85.09       | 85.74% | 5.7        | 83.68 | 14.69% | 1.3        |
> | Photo      | **86.64**       | 97.83% | 14.6       | 73.86 | 22.75% | 3.1        |
> | \# Average | 76.25       | 89.38% | 19.95      | 70.86 | **21.20%** | **3.4**        |
>
> | BERT-Base       || Co-Training         |            |   |  TCL       |            |
> | ---------- | ----------- | ------ | ---------- | ----- | ------ | ---------- |
> |            | **Acc**         | **Memory** | **Time (min)** | **Acc**   | **Memory** | **Time (min)** |
> | Arxiv      | \-          | OOM    | \-         | **74.87** | 70.73% | 130        |
> | Children   |  -          | OOM    | \-         | **60.73** | 80.99% | 30         |
> | History    | \-          | OOM    | \-         | **86.80**  | 98.73% | 18         |
> | Photo      | \-          | OOM    | \-         | 82.85 | 70.65% | 120        |
> | \# Average | \-          | OOM    | \-         | **76.31** | 80.28% | 74.50      |

---

> ### Author Response · Authors · 2023-08-24
> **Response to reviewer ghFj (Part 2)**
>
> **Limitation**:
>
> >Q3:  Many details are in appendix, which is absolutely fine, as long as the authors indicate clearly where to find additional details via coreference. Many of these coreferences are missing.
>
> A3: We appreciate your thorough suggestions. we have incorporated the suggested modifications into the manuscript, specifically refining Section 4.1, Section 4.4, and other relevant sections as per your insights. Furthermore, we have undertaken a thorough review of the remaining portions of the document to ensure precise cross-referencing, effectively guiding the reader to the accurate appendix sections.
>
> >Q4: Code documentation. This is not clear from the codebase how to run the benchmark, the authors need to specify scripts to call and also add a dir structure, which gives an overview of the scripts.
>
> A4: Thanks for your helpful suggestion. To enhance the accessibility of our project, we have implemented a comprehensive directory structure, enabling researchers to navigate and engage with the project more effectively. Please refer to Github:(https://github.com/sktsherlock/TAG-Benchmark) for the details.
>
> **Correctness**:
>
> >Q5:  l.114: "three-level label" for each book, does the node classification classify the hierarchy as well, or they are flattened, or just the lowest level of labels? Same question for l.122 on dataset Ele-Computer/Photo.
>
> A5: Thanks for raising the question. The node classification is defined to classify the lowest-level labels in the category hierarchy. For example, in the 'Books-History' dataset, the labels of each book used for the classification task are  'Europe', 'Asia', 'Africa' and so on.  Similarly, we classify the lowest-level labels in Ele-Computer (e.g., 'Data Storage' and 'Computer Components') and Ele-Photo datasets (e.g., 'Film Photography' and 'Digital Cameras').
>
> >Q6: The authors mention in Section 4.3 that the performance of TMDC is bad on the dataset Photo because the text on each node is the review. Can we use the product description as node text attributes, or that is not an option?
>
> A6: Thank you for bringing up this question. When developing the text-attributed graph datasets, we tried to incorporate various types of node text attributes. This approach allows us to create diverse datasets that simulate the variability of real-world data. Regarding the Ele-Photo and Ele-Computers datasets, we incorporate the user reviews as node attributes in the manuscript. This choice is grounded in two key rationales. First, it contributes to investigating the effect of different types of node attributes. Second, it aligns with the construction methodology commonly employed in the widely recognized Amazon-Computer and Amazon-Photo datasets within the GNN community [1]. This strategy advances the transferability of pertinent insights and methodologies derived from earlier datasets. Following your valuable suggestion, we have extended our experimentation to include product descriptions as well. In the following table, datasets labeled with 'RW' incorporate user reviews as node attributes, while those marked 'DS' employ product descriptions as attributes. Different training paradigms are systematically evaluated under the node classification task. BERT-Base is selected as the foundational text encoder model. Evidently, the performance of PLMs when utilizing descriptions as node attributes demonstrates a substantial enhancement in contrast to reviews. This observation underscores the pivotal role that the selection of node attributes plays in achieving desirable TAG representation learning. We have added the detailed discussions in Appendix D.9.
>
> |      |      | Photo-RW | Photo-DS | Computers-RW | Computers-DS |
> | ---- | ---- | -------- | -------- | ------------ | ------------ |
> | **LMs**  | PLM  | 77.53    | 85.07    | 61.96        | 86.41        |
> | **GNNs** | SAGE | 83.27    | 84.86    | 88.30         | 88.90         |
> |      | GCN  | 82.70     | 83.72    | 87.86        | 88.26        |
> |      | GAT  | 83.74    | 85.16    | **88.63**        | 89.13        |
> | **TPT**  | TMLM | 78.54    | 86.15    | 63.43        | 87.56        |
> |      | TDK  | 81.04    | 86.49    | 82.85        | 87.26        |
> |      | TCL  | 82.85    | 87.26    | 64.93        | 88.96        |
> |      | TMDC | **84.09**    | **88.15**    | 69.45        | **89.53**        |
>
> [1] Pitfalls of graph neural network evaluation. CoRR abs/1811.05868
>
> > Q7: Same question for Sports-Fitness, what is the text attribute?
>
> A7: Thanks for this question.  The text attribute for each node in the Sports-Fitness dataset is the title of the item, e.g., "Girls Ballet Tutu Neon Orange".

---

> > ### Author Response · Authors · 2023-08-24
> > **Response to reviewer ghFj (Part 3)**
> >
> > **Correctness**:
> >
> > >Q8: Why two different metrics for CitationV8 and GoodReads? MRR and Hits@K, why not evaluate both for these two datasets?
> >
> > A8: Thanks for your helpful suggestion. The link prediction task on the GoodReads dataset is similar to the real-world recommendation task, which focuses more on the accurate selection of top-k candidates. Thus, following previous works [1], we select the Hits@K metric. Conversely, when processing the CitationV8 dataset, we primarily follow the evaluation criteria (i.e., MRR) established on the ogbl-citation2 dataset [2] to evaluate the ranking list.
> > We totally agree that evaluating both datasets using MRR and Hits@K offers a more holistic assessment of performance.  We have accordingly revised the experimental results as follows. More detailed experimental results can be found in Appendix D.2 of the revised version.
> >
> > | CitationV8 |           | Hits@10 | Hits@50 | Hits@100 | MRR           |
> > | ---------- | --------- | ------------------------ | ------------------------ | -------------------------- | ------------- |
> > | PLM        | BERT-Tiny | 33.56 ± 1.56             | 48.15 ± 2.02             | 66.56 ± 0.56               | 41.23 ± 1.39  |
> > |            | BERT-Base | 38.86 ± 2.53             | 57.53 ± 1.96             | 72.44 ± 0.98               | 44.56 ± 1.23  |
> > | GNN        | T-GCN     | 50.89 ± 3.56             | 74.26 ± 2.16             | 90.23 ± 0.89               | 60.79 ± 0.28  |
> > |            | T-SAGE    | 45.12 ± 3.26             | 66.23 ± 1.56             | 89.36 ± 0.99               | 54.64 ± 1.03  |
> > |            | B-GCN     | 50.39 ± 4.56             | 75.12 ± 2.56             | 90.16 ± 0.46               | 60.04 ± 0.86  |
> > |            | B-SAGE    | 44.12 ± 4.12             | 71.26 ± 1.67             | 89.12 ± 0.75               | 53.96 ± 1.24  |
> > | TCL        | BERT-Tiny | 41.26 ± 1.49             | 57.26 ± 1.59             | 72.62 ± 0.96               | 47.26 ± 1.23  |
> > |            | BERT-Base | 46.58 ± 1.69             | 65.77 ± 2.01             | 72.46 ± 0.56               | 52.78 ± 2.03  |
> > |            | T-GCN     | 65.23 ± 2.36             | 81.23 ± 1.36             | 92.56 ± 0.56               | 65.69 ± 0.42  |
> > |            | T-SAGE    | 61.89 ± 3.21             | 80.12 ± 1.46             | 90.23 ± 0.89               | 55.70 ± 0.15  |
> > |            | B-GCN     | **68.26 ± 2.66**             | **84.56 ± 0.56**             | **93.68 ± 0.26**               | **70.16 ± 0.38**  |
> > |            | B-SAGE    | 62.36 ± 3.46             | 80.56 ± 1.56             | 92.56 ± 0.68               | 60.18  ± 0.15 |
> >
> > | GoodReads |           | Hits@10 | Hits@50 | Hits@100 | MRR          |
> > | --------- | --------- | ------------------------ | ------------------------ | -------------------------- | ------------ |
> > | PLM       | BERT-Tiny | 36.86 ± 2.04             | 52.45 ± 1.69             | 76.23 ± 1.11               | 42.15 ± 0.86 |
> > |           | BERT-Base | 43.96 ± 2.26             | 60.87 ± 1.43             | 79.22 ± 0.46               | 44.43 ± 1.15 |
> > | GNN       | T-GCN     | 61.47 ± 4.65             | 84.14 ± 2.15             | 90.43 ± 0.60               | 69.44 ± 0.56 |
> > |           | T-SAGE    | 64.52 ± 3.18             | 82.65 ± 1.45             | 88.53 ± 0.61               | 74.36 ± 0.84 |
> > |           | B-GCN     |  55.42 ± 5.83            | 85.04 ± 2.34             | 91.49 ± 1.25               | 65.12 ± 0.48 |
> > |           | B-SAGE    | 54.05 ± 3.03             | 82.87 ± 0.89             | 89.61 ± 0.33               | 65.68 ± 1.13 |
> > | TCL       | BERT-Tiny | 45.47 ± 2.53             | 61.56 ± 1.56             | 82.43 ± 0.49               | 55.12 ± 0.84 |
> > |           | BERT-Base | 52.59 ± 2.27             | 65.97 ± 1.23             | 85.56 ± 0.34               | 61.21 ± 1.23 |
> > |           | T-GCN     | 69.58 ± 2.29             | 88.89 ± 0.56             | 93.12 ±  0.57              | 83.16 ± 1.04 |
> > |           | T-SAGE    | 70.28 ±  2.36            | 85.12 ±  0.47            | 90.38 ±  0.57              | 81.12 ± 1.23 |
> > |           | B-GCN     | **73.87 ± 3.36**             | **92.82 ± 0.16**             | **95.85 ± 0.15**               | **85.12 ± 1.56** |
> > |           | B-SAGE    | 75.16 ± 2.26             | 90.74 ± 0.16             | 94.01 ± 0.11               | 82.15 ± 1.15 |
> >
> > [1] HOP-rec: high-order proximity for implicit recommendation. RecSys 2018
> >
> > [2] Open Graph Benchmark: Datasets for Machine Learning on Graphs. NeurIPS 2020.
> >
> > >Q9: Negative samples in the CitationV8 and GoodReads datasets.
> >
> > A9: Thank you for your inquiry. Negative samples refer to the nodes that are not directly connected to the target node. For each node, we generate 2k (or 5k) negative samples for the CitationV8  (or GoodReads) dataset. To facilitate a fair comparison, we sample the negative samples for each node during the dataset construction process. This pipeline aligns with the established practice for link prediction tasks in the OGB (Open Graph Benchmark).

---

> ### Author Response · Authors · 2023-08-24
> **Response to reviewer ghFj (Part 4)**
>
> **Clarity about Text**:
>
> >Q10:   General remark about footnotes. They should always come after punctuation and without spacing in between, e.g, line 188.
>
> A10: Thank you for bringing the mistake to our attention. We have rectified it in the updated version.
>
> **Clarity about Table:**
>
> >Q11:  Table 1: Add the tasks of node classification and link predictions to the datasets.
>
>  A11: Thanks for your suggestions and we have updated Table 1 in the revised version.
>
> >Q12: Write the tables in the format of Table X: line 277, (Table 2), line 301 (Table 4).
>
> A12: Thanks for your suggestions. We have corrected the format of the tables in the revised version.
>
> >Q13:  Report results like Table 2 for all the datasets in a bigger table. For instance, you can list some important findings with numbers in the main text and leave the big table to appendix. The current structure makes it hard to find all the results in one place. I find it hard to jump between tables and compare the performances.
>
> A13: Thanks for your suggestions. We have added a large table for each dataset in Appendix D.1, which contains all the results of different training paradigms.
>
> >Q14:  Table 2: Any reasons why other PLM-based methods like RoBERTa are not tested? Make sure you indicate what "scale" means here (in the appendix for instance), as it might not be clear to all readers that "scale" means the different versions of PLMs (number of parameters). Please clarify what "diff" means. It is the between the best and worst performers for one model (column). I would suggest to also mention the bold numbers are for each model (row)
>
> A14: Thanks for pointing out these concerns. Table 2 in the manuscript focuses on the impact of different PLM-based node features on the performance of GNNs, therefore we do not report the pure PLM-based results in Table 2. Following your suggestions, we have added a large table for each dataset to present the performance of all methods in Appendix D.1.  We have added more explanations on the definition of 'scale' in Appendix A.1, which is related to the number of parameters of different LMs.
> We also add the definition of 'diff' in the title of tables, which denotes the performance gap between the best and worst performers.
> Following your suggestion, we mark the best performer in each row with blue bold font and mark the best performer in each column with black bold font.
>
> > Q15:  Table 3: Maybe it would be clearer to mark the highest number per row (dataset) bold and mark boundaries between PLM based, GNN based and Co-training based. Also for me, it is counterintuitive to mark the increase with red and decrease with green.
>
> A15: Thanks for your suggestion. We have updated Table 3 by adding more colored marks in the revised version.

---

> > ### Author Response · Authors · 2023-08-24
> > **Response to reviewer ghFj (Part 5)**
> >
> > **Clarity about Table:**
> >
> > > Q16: Table 4: It makes sense to add a vertical line between the datasets. There are only four datasets shown here, what about the other two on node classification?
> >
> > A16: Thanks for your valuable suggestion. We've now included the vertical lines to demarcate the datasets. Owing to space limitations, we've relocated the results of the remaining two node classification experiments to Appendix D.1. Please refer to Appendix D.1 for the detailed results. We have emphasized these results more explicitly in the revised version.
> >
> >
> > >Q16.1: If I am not mistaken, 84.15 (Large-RoBERTa-Photo-GNNs) is the best performer on the dataset Photo, which is still worse compared to 86.64 (Photo-Co-Training based) in Table 3.
> >
> > A16.1: Sorry for the misunderstanding. 84.15 is the best result among all PLM-based, GNN-based and topology pre-training-based methods on the Photo dataset. Table 3 does not report the performance of co-training methods. In order to provide a comprehensive overview of all methods, we have added a large table for each dataset in Appendix D.1. Sorry again for the inconvenience caused by the table design.
> >
> > >Q16.2: I would suggest the authors to rethink the table strutures of Tables 2, 3, 4. It would make more sense to put them into one big table as the results are interlinked and it is easier to compare.
> >
> > A16.2: We sincerely appreciate your effort in improving the quality of our paper. A number of methods and their combinations are extensively evaluated, including PLM-based, GNN-based, co-training-based and topological pretraining-based ones.
> > The detailed models include 5 PLMs with different scales, 9 GNNs with node attribute vectors learned by different PLMs, co-training BERT-Tiny with individual GNNs, and different PLMs employed with four topological pre-training tasks. It is non-trivial to demonstrate all results in the main body.
> > Hence, in the previous version, we organized all results into distinct tables, each serving a specific purpose: e.g., analyzing the effects of different PLMs' features on the downstream GNNs through Table 2, analyzing the advantages and disadvantages of co-training methods through Table 3 and Figure 4; and demonstrating the feasibility of the topological pre-training strategies through Table 4. We totally agree with you that a comprehensive table encompassing all results is indeed more suitable. Such large tables have been added in Appendix D.1 for the overview of all evaluations.
> >
> > >Q16.3: If you want to keep the structure like Table 4, please at least mention that the GNN column is the best performers of GNN methods, and you can do the same for PLMs (add a PLM column) and co-training (add a co-training column and report the best GNN variant for each PLM variant).
> >
> > A16.3: Following your previous constructive comment, we have added the overall tables in Appendix D.1.
> > Meanwhile, we also updated Table 4 in the revised version accordingly.
> >
> > >Q16.4: The number in subscript is computed wrt the lowest number in row, please mention that and rethink the coloring.
> >
> > A16.4: Thanks for your suggestion. We have added the meaning of numbers in subscript in the title. We remove the colors for clarity.
> >
> > >Q16.5: Also it would be interesting to comment on the columns, in general, the larger the scale of an LLM is, the better the performance is. But it is not always the case, see column (History-GNNs), the best performer is Small-DistilBERT, 85.56.
> >
> > A16.5: Thank you for taking the time to carefully review our paper. In experiments like History-GNN, the PLMs primarily function as text encoding models, generating initial node features for the subsequent GNNs. It's important to note that the training of the downstream GNNs doesn't back-propagate the gradients to optimize the PLMs. Consequently, the performance of downstream GNNs isn't directly linked to the scale of the PLMs. While it's generally true that larger PLMs tend to possess enhanced learning capacities, this principle might not hold in the aforementioned scenario.
> > The static node embeddings generated by LMs are not inherently guaranteed to be relevant to the downstream graph-based tasks such as node classification and link prediction.
> > Moreover, DistilBERT is distilled from the BERT-Base model with more parameters, which is expected to preserve the knowledge within the teacher BERT model. Thus, its performance seems to be reasonable. We have added more discussion in the revision.

---

> > > ### Author Response · Authors · 2023-08-24
> > > **Response to reviewer ghFj (Part 6)**
> > >
> > > **Clarity about Figure**:
> > >
> > > >Q17: Figure 2: There are eight datasets, of which we only see 6 points. Please clarify.
> > >
> > > A17: Thank you for your thorough reading and consideration. The missing two datasets (GoodReads and CitationV8) are much larger than the other ones. If we put all datasets in the same coordinate system, there will be huge empty spaces between these two datasets and other ones, leading to inferior reading experiences. We have clarified this issue in the revised version.
> > >
> > > >Q18: Figure 4: It is not clear what "Fanout" means. But I guess it is supposed to be the number of neighbors in the x-axis. Please add markers to the lines, as the colors are not black-white-print friendly.
> > >
> > > A18: Thanks for your valuable suggestion. 'Fanout' denotes the number of sampled neighboring nodes to which the center node is directly connected during the training of GNNs. We have added explanations in the caption of Figure 4. Meanwhile, we have added markers to the lines in Figure 4 for reading friendly.
> > >
> > > **Documentation**:
> > >
> > > >Q19: Code documentation. This is not clear from the codebase how to run the benchmark, the authors need to specify scripts to call and also add a dir struture, which gives an overview of the scripts.
> > >
> > > A19: Thanks for pointing out these concerns. We have added scripting instructions to the codebase to facilitate the researcher's experiments. We also add a new figure to illustrate the code structure of our project.  For a comprehensive walkthrough of the training process, a detailed set of instructions is available at the following Github:(https://github.com/sktsherlock/TAG-Benchmark).
> > >
> > > **Additional Feedback:**
> > >
> > > >Q20:  Double check the formatting, e.g., l. 310, Section 4.4 should not end with a dot (.), apart from the comments in the "Clarity" section.
> > >
> > > A20: Thanks for pointing out this problem. We have double-checked the formatting and corrected these mistakes accordingly in the revised version.
> > >
> > > >Q21: I briefly went over the appendix and found many details. Some corrections:
> > > 	1) Section A.2, $ℎ$ is used in the main text, and $z$ in appendix, please make them consistent.
> > > 	2) I have suggested merging the tables into a bigger one, also consider doing so for Tables 3-16, based on various metrics, of course.
> > > 	3) Any reason why a table like Table 3 is not produced for the link prediction task?
> > >
> > > A21: Thank you for your thoughtful suggestions.
> > >
> > > a) We have addressed the issue of symbol inconsistency in the revised version accordingly.
> > >
> > > b) In Appendix D.1, Tables 3-16 are restructured and combined into larger tables as Table 3-5 in the revised Appendix D.1. Results of different training paradigms on each dataset are reported within a single table. This modification offers a comprehensive comparison of various training approaches under different metrics, enhancing the reading experience.
> > >
> > > c) We have added Table 6 in Appendix D.2 to demonstrate the performance of various models under the link prediction task. One can see that the GNN-based methods consistently outperform the LMs, demonstrating that topology structures are more significant in the link prediction task.

---

> > > > ### Comment · Reviewer_ghFj · 2023-08-27
> > > >
> > > > Thank you very much for making the changes in presenting the results. The writing quality has substantially improved.

---

> > > > > ### Author Response · Authors · 2023-08-28
> > > > >
> > > > > Dear reviewer ghFj,
> > > > >
> > > > > Thanks so much for your hard work in reading and reviewing our paper. All your comments are really helpful to improve our paper, and we appreciate them.
> > > > >
> > > > > Sincerely,
> > > > >
> > > > > Authors

---

### Official Review · Reviewer_pXJE · 2023-07-21
**A Comprehensive Study on Text-attributed Graphs: Benchmarking and Rethinking**

**Rating:** 6
**Confidence:** 5
**Clarity:** The paper is well written and easy to…

**Strengths:**

1.	This paper presents a comprehensive and diverse collection of challenging benchmark datasets for text-attributed graphs, with the potential to inspire further research in this direction.
2.	The detailed information regarding data collection and their statistics is well-explained.
3.	The authors also propose several warm-up methods to explore the topology structure in LLMs. They conduct a series of experiments based on these methods to benchmark their results on the proposed datasets.


**Additional Feedback:**

Please see the Opportunities for Improvement section for details.

**Correctness:**

The datasets are collected in a sound way. It also provides a preliminary benchmark for existing methods.



**Documentation:**

Yes

**Opportunities For Improvement:**

1.	The experiments could be further improved by including traditional supervised GNN models as strong baselines. Following previous protocols, these supervised GNNs could take the encoded document embeddings obtained from shallow embedding models like Skip-gram and GloVe as input.
2.	While the main classification results in the paper focus on the supervised setting, it would be beneficial to evaluate the warm-up methods under semi-supervised or few-shot scenarios as well. This would provide a more comprehensive evaluation and deeper insights into their performance.
3.	The connection to previous work should be thoroughly discussed. For instance, it is essential to highlight the crucial differences between the Arxiv-TA dataset and the original ogbn-Arxiv. Given that the original ogbn-Arxiv dataset also provides associated documents, it might be more appropriate to retain its original name for better clarity and comparison.


**Relation To Prior Work:**

Yes

**Summary And Contributions:**

Pre-trained language models (PLMs) have demonstrated significant potential in natural language and computer vision domains. Motivated by this success, there is a growing interest in harnessing PLMs to enhance graph neural networks (GNNs) for text-attributed graphs (TAGs). However, the lack of benchmark datasets and standardized evaluation procedures specifically tailored for TAGs has hindered advancements in this area. To address this gap, this paper introduces a new set of TAGs datasets and conducts a comprehensive series of experiments across various learning paradigms to explore the capabilities of PLMs on graphs.

---

> ### Author Response · Authors · 2023-08-24
> **Response to reviewer pXJE (Part 1)**
>
> >Q1: The experiments could be further improved by including traditional supervised GNN models as strong baselines. Following previous protocols, these supervised GNNs could take the encoded document embeddings obtained from shallow embedding models like Skip-gram and GloVe as input.
>
> Thank you for your insightful suggestion. Our manuscript incorporates a total of seven Graph Neural Networks (GNNs), specifically GCN, SAGE, GAT, GIN, JKNet, APPNP, and MoNet. To establish the supervised GNN baselines, we first employ Pre-trained Language Models (PLMs) to encode textual content within nodes into static embeddings. Subsequently, we employ the aforementioned GNNs to aggregate neighborhood information. The gradients from downstream tasks, such as node classification, are then back-propagated to update GNN parameters. Table 2 in the manuscript presents the performance of various supervised GNNs.
>
> Building on your valuable feedback, we have introduced two additional recent GNNs, RevGAT[1] and NodeFormer[2], facilitating a more comprehensive comparative analysis. Moreover, we have extended our node text encoding approach by incorporating shallow models like Skip-Gram[3] and GloVe[4]. Experimental results (accuracy) on node classification tasks over multiple datasets are presented in the following table. One can clearly see that GNNs equipped with deeper text encoders consistently outperform those with shallow encoders, verifying the importance of node attribute understanding. We have added the detailed results and discussions in the revised Appendix D.4.
>
>
> | Arxiv         |      | GCN   | GAT   | SAGE  | RevGAT | NodeFormer |
> | ------- | ---------- | ----- | ----- | ----- | ------ | ---------- |
> | Shallow | Skip-Gram  | 71.97 | 72.22 | 72.02 | 73.42  | 71.03      |
> |         | GloVe      | 72.12 | 72.54 | 72.48 | 73.51  | 72.04      |
> | PLMs    | DistilBERT | 73.39 | **73.48** | 74.48 | 74.68  | **73.56**     |
> |         | BERT       | 73.30  | 73.40  | 74.14 | 74.59  | 72.80      |
> |         | RoBERTa    | **73.56** | 73.38 | **74.52** | **74.82**  | 73.12      |
>
> |  Children       |              | GCN   | GAT   | SAGE  | RevGAT | NodeFormer |
> | ------- | ------------ | ----- | ----- | ----- | ------ | ------- |
> | Shallow | Skip-Gram    | 56.23 | 55.84 | 56.49 | 57.13  | 56.03   |
> |         | GloVe        | 57.02 | 56.58 | 57.22 | 58.12  | 56.86   |
> | PLMs    | DistilBERT   | 58.19 | 57.91 | **59.33** | **59.28**  | **58.03**   |
> |         | BERT-Base    | 58.11 | 57.70 | 58.74 | 58.67  | 57.42   |
> |         | RoBERTa-Base | **58.62** | **57.83** | 58.97 | **59.01**  | 57.26   |
>
> [1] Training Graph Neural Networks with 1000 Layers.  ICML 2021
>
> [2] NodeFormer: A Scalable Graph Structure Learning Transformer for Node Classification. NeurIPS 2022
>
> [3] Efficient Estimation of Word Representations in Vector Space. ICLR 2013
>
> [4] GloVe: Global Vectors for Word Representation. EMNLP 2014

---

> > ### Author Response · Authors · 2023-08-24
> > **Response to reviewer pXJE (Part 2)**
> >
> > > Q2: While the main classification results in the paper focus on the supervised setting, it would be beneficial to evaluate the warm-up methods under semi-supervised or few-shot scenarios as well. This would provide a more comprehensive evaluation and deeper insights into their performance.
> >
> > Your constructive suggestion is greatly appreciated. We fully concur that assessing warm-up methods (i.e., "topological pre-training" abbreviated as TPT) significantly enhances the depth of comprehension and insight into such pre-training tasks. Taking your suggestion into account, we've incorporated two distinct updates, which have been provided in the revised Appendix D.5.
> > 1) To effectively evaluate the performance of various methods within semi-supervised settings, we've adjusted the training ratio from 20\% to 100\%. 20\% implies that only 20\% of training samples are used in the model training process. BERT-Base is selected as the foundational text encoder model. Detailed experimental results concerning the node classification task across four datasets have been tabulated below. Notably, as the training ratio decreases, the benefits of topological pre-training approaches become even more significant. Among these strategies, TMDC demonstrates superior performance across all datasets within the semi-supervised context.
> >
> >
> > | Arxiv    |  | 20%   | 40%   | 60%   | 80%   | 100%  |
> > | --- | ----- | ----- | ----- | ----- | ----- | ----- |
> > | **LMs** | PLM   | 57.76 | 62.56 | 67.12 | 70.15 | 72.96 |
> > | **TPT** | TMLM  | 62.15 | 65.51 | 69.01 | 71.76 | 73.97 |
> > |     | TDK   | 63.86 | 66.78 | 70.01 | 71.95 | 74.23 |
> > |     | TCL   | 64.98 | 67.89 | 71.22 | 73.56 | 74.87 |
> > |     | TMDC  | **65.48** | **68.91** | **72.92** | **75.09** | **76.11** |
> >
> > | History |  | 20%   | 40%   | 60%   | 80%   | 100%  |
> > | ---- | ------- | ----- | ----- | ----- | ----- | ----- |
> > | **LMs**  | PLM     | 70.86 | 75.61 | 80.18 | 83.28 | 86.09 |
> > | **TPT**  | TMLM    | 74.39 | 77.70  | 81.21 | 84.03 | 86.24 |
> > |      | TDK     | 75.46 | 78.93 | 82.17 | 84.18 | 86.46 |
> > |      | TCL     | 77.12 | 80.53 | 83.46 | 85.49 | 86.80  |
> > |      | TMDC    | **78.24** | **81.48** | **84.23** | **85.80**  | **86.82** |
> >
> > | Children |  | 20%   | 40%   | 60%   | 80%   | 100%  |
> > | ---- | -------- | ----- | ----- | ----- | ----- | ----- |
> > | **LMs**  | PLM      | 44.21 | 49.12 | 53.8  | 56.97 | 59.91 |
> > | **TPT**  | TMLM     | 48.52 | 51.88 | 55.38 | 58.13 | 60.34 |
> > |      | TDK      | 49.46 | 52.98 | 56.21 | 58.15 | 60.43 |
> > |      | TCL      | 51.21 | 54.67 | 57.59 | 59.55 | 60.73 |
> > |      | TMDC     | **52.88** | **56.17** | **58.91** | **60.41** | **61.43** |
> >
> > | Photo |  | 20%   | 40%   | 60%   | 80%   | 100%  |
> > | ---- | ----- | ----- | ----- | ----- | ----- | ----- |
> > | **LMs**  | PLM   | 56.61 | 63.28 | 69.45 | 73.86 | 77.53 |
> > | **TPT**  | TMLM  | 66.72 | 70.08 | 73.58 | 76.33 | 78.54 |
> > |      | TDK   | 70.07 | 73.59 | 76.82 | 78.76 | 81.04 |
> > |      | TCL   | 73.20  | 76.66 | 79.58 | 81.54 | 82.85 |
> > |      | TMDC  | **75.54** | **78.83** | **81.57** | **83.07** | **84.09** |
> >
> > 2) Subsequently, we've undertaken few-shot learning experiments over four datasets. The term 'K-shot' denotes that merely K samples correspond to each category within the training set. Based on the results in the following tables, topological pre-training consistently enhances the performance of LMs across diverse few-shot scenarios.
> >
> > | 3-shot  |  | Arxiv | Children | History | Sports |
> > | ---- | ------ | ----- | -------- | ------- | ------ |
> > | **LMs**  | PLM    | 37.76 | 26.55    | 32.52   | 42.56  |
> > | **TPT**  | TMLM   | 40.08 | 31.86    | 35.51   | 44.18  |
> > |      | TDK    | 41.15 | 34.56    | 36.58   | 45.69  |
> > |      | TCL    | 43.26 | 38.26    | 38.95   | 48.58  |
> > |      | TMDC   | **45.68** | **40.05**    | **40.86**   | **50.26**  |
> >
> > | 5-shot |  | Arxiv | Children | History | Sports |
> > | ---- | ------ | ----- | -------- | ------- | ------ |
> > | **LMs**  | PLM    | 41.56 | 30.15    | 37.78   | 46.22  |
> > | **TPT**  | TMLM   | 45.21 | 36.89    | 40.15   | 50.15  |
> > |      | TDK    | 47.42 | 39.26    | 42.26   | 52.68  |
> > |      | TCL    | 49.58 | 42.12    | 44.12   | 55.64  |
> > |      | TMDC   | **51.52** | **44.62**    | **46.69**   | **58.95**  |

---

> ### Author Response · Authors · 2023-08-24
> **Response to reviewer pXJE (Part 3)**
>
> > Q3: The connection to previous work should be thoroughly discussed. For instance, it is essential to highlight the crucial differences between the Arxiv-TA dataset and the original ogbn-Arxiv. Given that the original ogbn-Arxiv dataset also provides associated documents, it might be more appropriate to retain its original name for better clarity and comparison.
>
> We appreciate your valuable suggestion. The primary difference between our research and the ogbn-arxiv dataset are twofold. First, ogbn-arxiv solely provides node embeddings acquired through shallow text encoders like Skip-Gram. This dataset does not delve into the performance of employing multiple PLM features in downstream GNNs, nor does it explore the co-training paradigm or topological pre-training techniques. Second, the original ogbn-arxiv dataset suffers from noises, including gibberish and missing attributes. In our utilized dataset, we have cleaned such instances of noisy and missing data. Considering the ogbn-arxiv is the basis of the utilized dataset, we have changed the title to ogbn-arxiv-TA (where "TA" denotes "text attribute") in the revised version. This alteration is intended to underscore the connection between our work and the ogbn-arxiv dataset, in line with your suggestion.
>
> To encapsulate, we've observed the prevalence of text-attributed graphs within previous graph datasets where nodes are enriched with substantial text attributes. Despite this, prevailing research within the graph domain has largely underestimated the significance of text attribute modeling. Numerous datasets fall short of providing raw node texts, instead relying on simplistic models like bag-of-words for encoding text attributes [1, 2]. Consequently, the initial node features derived from these approaches tend to lack meaningful semantic context.
>
> Stepping in, the Open Graph Benchmark (OGB) has introduced datasets housing unprocessed text, exemplified by ogbn-arxiv and ogbn-papers100M. These datasets have adopted a more progressive approach by employing models like Skip-Gram for encoding text attributes. The OGB leaderboard unveils a multitude of novel model architectures, such as GIANT[3] and GLEM[4], spearheaded by researchers to tackle graph machine learning tasks inherent to this specific data category.
>
> Despite this vibrant research landscape, there remains a conspicuous absence of a comprehensive benchmark tailored to the nuances of text-attributed graphs. It is within this void that we introduce our benchmark, with the optimistic intent of pushing the boundaries of the field by providing a comprehensive benchmark for a deeper exploration of this domain. Our principal contributions encompass an extensive study of the significance of node attribute modeling and diverse attribute-topology training paradigms. We anticipate that these contributions will inspire future research endeavors in this domain.
>
> [1] Collective classification in network data. AI Mag. 29(3): 93-106 (2008)
>
> [2] Pitfalls of graph neural network evaluation. CoRR abs/1811.05868
>
> [3] Node Feature Extraction by Self-Supervised Multi-scale Neighborhood Prediction. ICLR 2022
>
> [4] Learning on Large-scale Text-attributed Graphs via Variational Inference. ICLR 2023

---

> > ### Author Response · Authors · 2023-08-28
> >
> > Dear reviewer pXJE,
> >
> > Thanks again for your insightful comments, which, we believe, are very important to improve our paper. In the rebuttal, we have tried to answer your questions one by one. If you have further questions, we are very happy to discuss them. Thanks.
> >
> > Sincerely,
> >
> > Authors

---

### Author Response · Authors · 2023-08-24
**Rebuttal by Authors**

We thank all the reviewers for their insightful and constructive feedback. We have made point-to-point response to the comments of each reviewer.

Moreover, we have revised the manuscript based on the valuable suggestions from reviewers.

Finally, we once again thank all reviewers for their insightful comments which are very helpful for improving the quality of our paper.

---

### Decision · Program_Chairs · 2023-09-22

**Decision:**

Accept (Poster)

**Comment:**

All reviewers are in favor of accepting, the paper has a clear champion among the reviewers, and the rebuttal seems quite comphrehensive.